# Sperm motility in mice with oligo-astheno-teratozoospermia restored by in vivo injection and electroporation of naked mRNA

Charline Vilpreux[1†], Paul Fourquin[1†], Guillaume Martinez[1,2], Magali Court[1], Florence Appaix[3], Jean Luc Duteyrat[4], Maxime Henry[5], Julien Vollaire[5], Camille Ayad[6], Altan Yavuz[6], Geneviève Chevalier[1], Lisa De Macedo[1], Sofia Andrade Rebelo[1], Edgar Del Llano[1], Célia Tebbakh[1,2], Zine Eddine Kherraf[1,2], Emeline Lambert[1], Sekou Ahmed Conté[1], Zeina Wehbe[1,2], Elsa Giordani[1], Veronique Josserand[5], Jacques Brocard[4], Charles Coutton[1,2], Bernard Verrier[6], Pierre F Ray[1,7], Corinne Loeuillet[1], Christophe Arnoult[1], Jessica Escoffier[1]*

[1]Université Grenoble Alpes, Inserm U1209, CNRS UMR 5309, Team Genetic, Epigenetic and Therapies of Infertility, Institute for Advanced Biosciences, Grenoble, France; [2]UM de Génétique Chromosomique, Hôpital Couple-Enfant, CHU Grenoble Alpes, Grenoble, France; [3]University Grenoble Alpes, INSERM U1209, CNRS UMR5309, Optical microscopy and cell imaging (MicroCell) facility, Institute for Advanced Biosciences, Grenoble, France; [4]Université Claude Bernard Lyon 1, CNRS UAR3444, Inserm US8, ENS de Lyon, SFR Biosciences, Lyon, France; [5]Université Grenoble Alpes, Inserm U1209, CNRS UMR 5309, plateforme Optimal, Institute for Advanced Biosciences, Grenoble, France; [6]Université Claude Bernard Lyon 1 - Laboratoire de Biologie Tissulaire et d'Ingénierie Thérapeutique, UMR 5305, Université Lyon 1, CNRS, IBCP, Lyon, France; [7]UM GI-DPI, CHU Grenoble Alpes, Grenoble, France

*For correspondence:
jessica.escoffier@univ-grenoble-alpes.fr

†These authors contributed equally to this work

Competing interest: The authors declare that no competing interests exist.

## eLife Assessment

This study reports an mRNA-based strategy for restoring sperm motility in a mouse model of monogenic male infertility. The work is technically innovative and potentially **valuable**, as it demonstrates feasibility of in vivo testicular mRNA delivery without genomic integration of foreign DNA. However, although partial recovery of sperm motility is supported, the evidence for meaningful restoration of fertility remains **incomplete**, with weak IVF outcomes and difficult-to-interpret ICSI results. In addition, mechanistic questions regarding the persistence of mRNA and the specificity of germ-cell targeting remain insufficiently resolved, limiting the strength of the authors' conclusions.

**Abstract** Oligo-astheno-teratozoospermia (OAT), a recurrent cause of male infertility, is the most frequent disorder of spermatogenesis with a predominantly genetic origin. Patients and mice bearing mutations in the *ARMC2* gene exhibit reduced sperm concentration, multiple morphological defects, and impaired motility, defining a canonical OAT phenotype. Intracytoplasmic sperm injection (ICSI) is required to treat this condition; however, it is associated with a slightly increased risk of birth defects compared with natural conception, highlighting the need for novel targeted therapies. Here, in vivo testicular injection followed by electroporation of capped, polyadenylated naked

messenger RNA (mRNA) was evaluated as a strategy to treat *ARMC2*-related infertility in mice. mRNAs encoding reporter proteins were used to assess expression efficiency and kinetics using in vivo and in vitro 2D and 3D imaging. Reporter proteins were detected in germ cells for up to three weeks, demonstrating the feasibility of mRNA-based approaches. These results were compared with a non-integrative plasmid Enhanced Episomal Vector, which induced weak and transient expression in spermatogenic cells. Delivery of *Armc2* mRNA restored morphologically normal and motile sperm in deficient males, capable of producing embryos via in vitro fertilization and ICSI. These findings provide proof-of-concept that mRNA electroporation can restore sperm motility and fertilizing potential, offering a novel strategy to correct monogenic male infertility.

## Introduction

Worldwide, 10–15% of couples (or 70 million) face infertility (*Boivin et al., 2007*). Infertility is thus a major public health issue presenting significant medical, scientific, and economic challenges (a multi-billion € annual market) (*Thonneau and Spira, 1991*). A significant proportion of infertility is due to altered gametogenesis, where the sperm and eggs produced are incompatible with fertilization and/or embryonic development (*Kekäläinen, 2021*). Approximately 40% of cases of infertility involve a male factor, either exclusively or associated with a female deficiency (*Kumar and Singh, 2015*).

Male gametogenesis, or spermatogenesis, is a highly complex physiological process that can be split into three successive steps: proliferation (mitosis of spermatogonia), reduction of the number of chromosomes (meiosis of spermatocytes), and morphogenesis of spermatozoa (spermiogenesis). Spermatogenesis involves the coordinated expression of a large number of genes, with approximately 2000 showing testis-enriched expression, about 60% of which are expressed exclusively in the testes (*Uhlén et al., 2016*). Because of this multiplicity of genes, spermatogenesis is strongly affected by genetic factors (*Uhlén et al., 2016*), with most severe disorders likely to be of genetic origin.

Among male infertility disorders, oligo-astheno-teratozoospermia (OAT) is the most frequent (50%; *Thonneau et al., 1991*), and it is likely to be of genetic origin. Spermatocytograms of OAT patients show a decrease in sperm concentration, multiple morphological defects, and defective motility (*Cavallini, 2006*; *Colpi et al., 2018*). Because of these combined defects, patients are infertile and can only conceive by intracytoplasmic sperm injection (ICSI).

ICSI can efficiently overcome the problems faced. Nevertheless, concerns persist regarding the potential risks associated with this technique, including blastogenesis defect, cardiovascular defect, gastrointestinal defect, musculoskeletal defect, orofacial defect, leukemia, central nervous system tumors, and solid tumors (*Hansen et al., 2005*; *Halliday et al., 2010*; *Davies et al., 2012*; *Kurinczuk et al., 2004*). Statistical analyses of birth records have demonstrated an elevated risk of birth defects, with a 30–40% increased likelihood in cases involving ICSI (*Hansen et al., 2005*; *Halliday et al., 2010*; *Davies et al., 2012*; *Kurinczuk et al., 2004*), and a prevalence of birth defects between 1% and 4% (*Davies et al., 2012*). It is important to note, however, that the origin of these risks remains debated. Several large epidemiological and mechanistic studies indicate that both the procedure itself (direct microinjection and in vitro manipulation) and the underlying genetic or epigenetic abnormalities often present in men requiring ICSI contribute to the observed outcomes (*Hansen et al., 2005*; *Davies et al., 2012*; *Graham et al., 2023*; *Palermo et al., 2017*). To overcome these drawbacks, a number of experimental strategies have been proposed to bypass ARTs and restore spermatogenesis and fertility, including gene therapy (*Usmani et al., 2013*; *Raina et al., 2015*; *Michaelis et al., 2014*; *Wang et al., 2022*).

Gene therapy consists of introducing a DNA sequence into the genome to compensate for a defective gene. It can thus rescue production of a missing protein and is now applied both in research (*Duan, 2024*) and for the treatment of human diseases (*Jacobson et al., 2021*).

Given the genetic basis of male infertility, the first strategy, tested in mice, was to overcome spermatogenic failure associated with monogenic diseases by delivery of an intact gene to deficient germ cells (*Usmani et al., 2013*). Gene therapy is effective in germ cells, as numerous publications have shown that conventional plasmids can be transferred into spermatogonia in several species with success, allowing their transcription in all cells of the germinal lineage (*Usmani et al., 2013*; *Raina et al., 2015*; *Michaelis et al., 2014*; *Wang et al., 2022*). Most experiments were performed in mouse models, delivering DNA constructs into living mouse germ cells by testis electroporation

after microinjection of a DNA-containing solution into the seminiferous tubules. Using this method, it was possible to rescue meiosis and fertility in mouse models of infertility (*Usmani et al., 2013*; *Wang et al., 2022*). However, the genetic changes induced are transmitted to any descendants. Consequently, gene therapy cannot be used to treat infertility in humans, both for ethical reasons and to avoid any eugenic deviations, and currently transmissible changes in humans are illegal in 39 countries (*Liu, 2020*). Furthermore, the genetic modification of germ cell lines poses biological risks, including the induction of cancer, off-target effects, and cell mosaicism. Errors in editing may have adverse effects on future generations. It is exceedingly challenging to anticipate the consequences of genetic mosaicism, for instance, in a single individual (*Sadelain et al., 2011*; *Ishii, 2017*). Gene therapies have thus raised both ethical controversy and long-term safety issues.

For these reasons, we decided to test an alternative strategy to DNA transfection based on the use of naked messenger RNA (mRNA). Thanks to this change, the risk of genomic insertion is avoided, and thus there is no question of heritable alterations (*Parhiz et al., 2024*). The first part of this study presents a characterization of the protein expression patterns obtained following transfection of naked mRNA coding for reporter genes into the testes of mice. The second part is to apply the protocol to a preclinical mouse model of OAT. Patients and mice carrying mutations in the *ARMC2* gene present a canonical OAT phenotype and are infertile. The preclinical *Armc2*-deficient (*Armc2* KO) mouse model is therefore a valuable model to assess whether in vivo injection of naked mRNA combined with electroporation can restore spermatogenesis. We chose this model for several reasons: first, *Armc2* KO mice are sterile and all sperm exhibit short, thick, or coiled flagella (*Graham et al., 2023*). As a result, 100% of sperm are immobile, thus it should be easy to determine the efficacy of the technique by measuring sperm motility with a computer-assisted semen analysis (CASA) system. Second, the *Armc2* gene codes for an 867-amino acid protein, and this large size represents a challenge for expression in the testis following electroporation.

To determine the efficacy of naked mRNA transfection as a method to achieve protein expression in the testes, we first assessed the level of transcription of reporter proteins after mRNA injection compared to the injection of a non-integrating plasmid, the Enhanced Episomal Vector (EEV). EEV is a vector derived from Epstein–Barr virus; it includes an origin of replication (EBV Ori) and the EBNA1 protein. Both elements allow the synchronous initiation of extra-chromosomal EEV replication with host DNA at each S phase of the cell cycle and the segregation of the EEV episome in daughter cells. It is notable that EEV is maintained at a rate of 90–95% per cell division. It does not integrate or modify the host genome (*Davies et al., 2012*; *Kurinczuk et al., 2004*).

In the present in vivo work, we injected and electroporated three distinct mRNAs coding for the following reporter proteins: GFP, luciferase (Luc), and mCherry, and an EEV episome vector containing the sequences coding for both GFP and luciferase reporter proteins. The initial step was to characterize and validate the method of injection in adult males. Subsequently, the kinetics and patterns of expression of the electroporated mRNAs and EEV were compared using a variety of methods, including whole testis imaging, in vivo bioluminescence imaging, and tissue clearing. Subsequently, the mRNA transfection protocol was tested in a preclinical mouse model of OAT with the objective of restoring fertility.

## Results

### In vivo microinjection and electroporation of mouse testes

Two routes have been described for microinjection of DNA into the testes: direct injection through the *tunica albuginea,* or injection into the lumen of the seminiferous tubules via the *rete testis*. We chose the *rete testis* route and evaluated the efficacy of the microinjection protocol. In particular, we wished to better characterize the diffusion of the injected solution in the volume of the testis, as we were unable to find any information on this parameter in the literature. The efficacy of microinjection *via rete testis* was assessed using fluorescent i-particles NIRFiP-180, and by measuring their diffusion in testis cross sections examined by microscopy 3 days post-injection (*Figure 1*). To avoid lesions due to overfilling, the injection was controlled by measuring the expansion of the staining of the peripheral seminiferous tubules during the injection. Injections were stopped when the testes were filled to 2/3 of their capacity (*Figure 1A, B*). In testis cross sections, the fluorescent i-particles NIRFiP-180 were heterogeneously distributed, and mainly observed in seminiferous tubules located in the peripheral

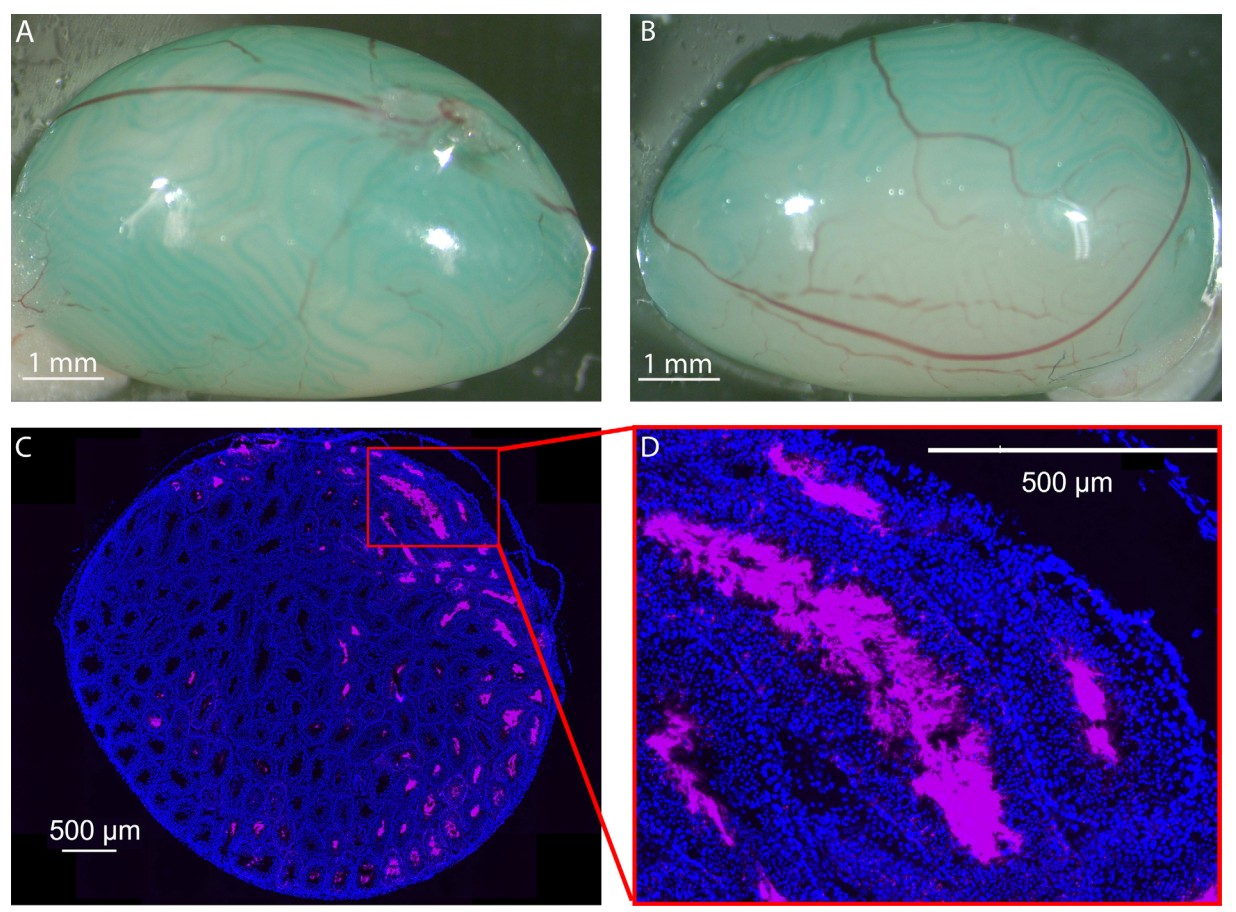

**Figure 1.** Distribution of i-particles NIRFIP-180 in testis injected via the rete testis route. (**A**) A solution containing 0.05% Fast Green and 1% fluorescent i-particles NIRFiP-180 was prepared, 10 μl was injected into the seminiferous tubules of adult males, through the *rete testes* and its efferent channels. Injection was performed at constant pressure under a binocular microscope. The progression of filling of the seminiferous tubules was monitored thanks to the Fast Green. (**B**) The testes were only filled to 2/3 capacity in order to prevent damage to the tissue. (**C**) Representative distribution of fluorescent i-particles NIRFiP-180 in a whole cross-section of an injected testis. Nuclei were counterstained with DAPI (blue emission) to reveal tubules. (**D**) Enlargement of a seminiferous tubule showing particles localized inside the lumens of the tubules. Scales bars: 1 mm and 500 μm.

The online version of this article includes the following source data and figure supplement(s) for figure 1:

**Figure supplement 1.** EEV and mRNA maps.

**Figure supplement 1—source data 1.** PDF file containing original agarose gel electrophoresis for *Figure 1—figure supplement 1D* indicating the relevant bands.

**Figure supplement 1—source data 2.** Original files for agarose gel electrophoresis analysis displayed in *Figure 1—figure supplement 1D*.

**Figure supplement 2.** Damaged tubules observed by optical microscopy following overstimulation.

region of the testes, with fewer particles present in the center of the testes (*Figure 1C, D*). Moreover, no fluorescent i-particles NIRFiP-180 were visible in the peritubular space. These results indicated that microinjection through the *rete testis* did not produce a homogenous distribution of the particles throughout the seminiferous tubules. Nevertheless, the seminiferous tubules remained intact, as no signal was observed in the peritubular space (*Figure 1C, D*).

Next, we assessed the overall safety of the *rete testis* microinjection and electroporation of mRNA and EEV into testes. The safety of the protocol was evaluated by comparing macroscopic and microscopic anatomies of control (injected with control solution, PBS, 0.05% FG), and treated testes (injected either with EEV-*GFP* (PBS, 0.05% FG) or *GFP*-mRNA (PBS, 0.05% FG)). Three days post-injection, the testes were first observed under a binocular microscope to identify possible macroscopic degeneration of the seminiferous tubules (*Figure 2A1, B1*). Degenerations appear as pearly white lesions at the surface of the testis as illustrated in *Figure 1—figure supplement 2* following

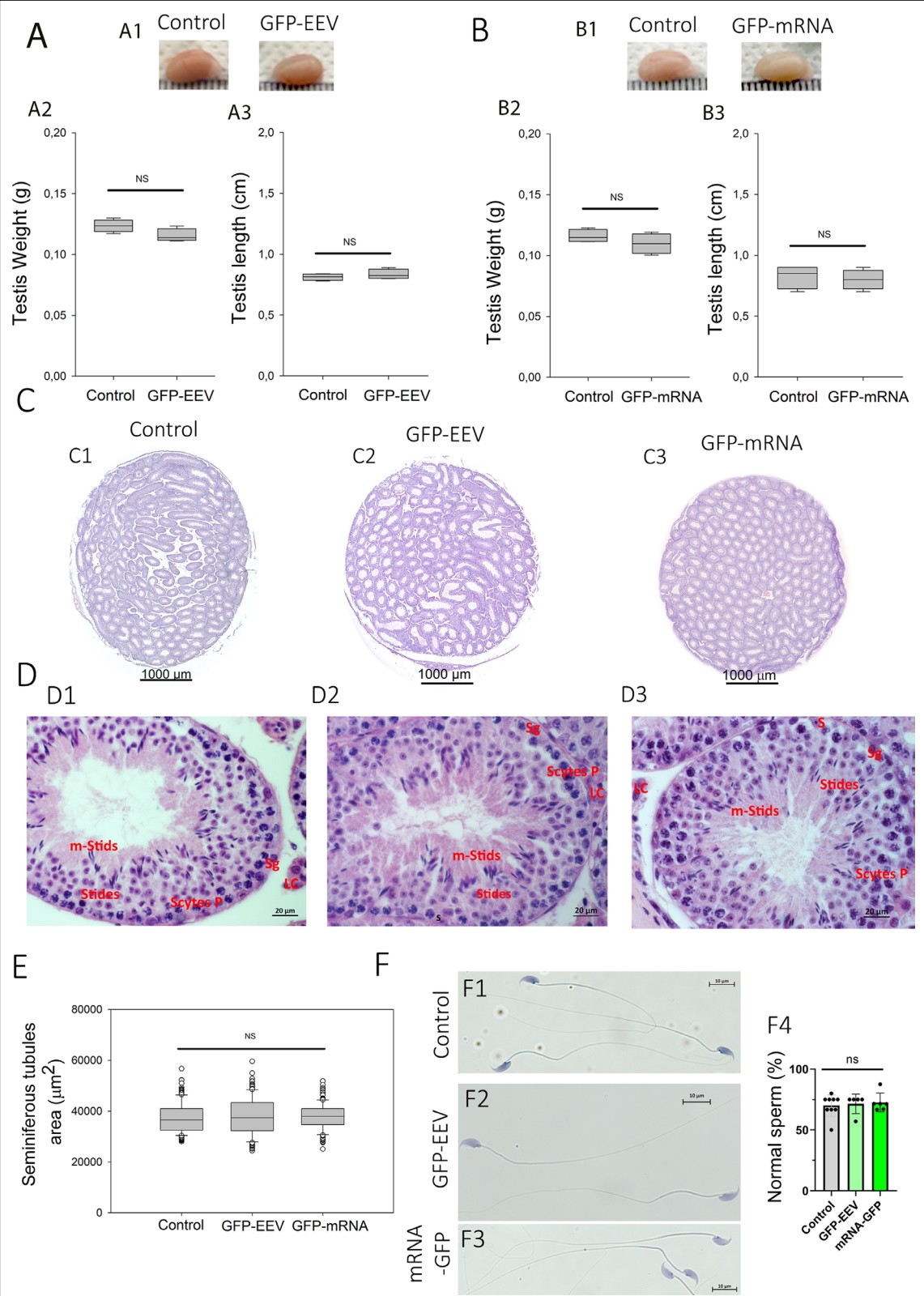

**Figure 2.** In vivo injection and electroporation do not alter the morphological structure of the testes, seminiferous tubules, or sperm cells. Testicular morphology was not affected by in vivo injection and electroporation of EEV-*GFP* (**A**) or *GFP*-mRNA (**B**). Controls correspond to contralateral testes injected/electroporated with control solution (PBS, 0.05% FG). (**A1, B1**) Comparison of the testicular morphology of adult testes injected with nucleic acid vectors or control solutions. (**A2, B2**) Comparison of testicular weight and (**A3, B3**) testicular length on day 7 after injection/electroporation. Data

*Figure 2 continued on next page*

*Figure 2 continued*

are represented as a box plot median (n = 4 for each condition). A Wilcoxon matched pairs test was used to assess the significance of any differences in testis weights and lengths, and p values of ≤0.05 were considered statistically significant. (**C**) Intact testicular structure after in vivo injection and electroporation with EEV-*GFP* and *GFP*-mRNA. Comparison of testicular cross-section structures. Testes paraffin sections were stained with eosin/hematoxylin and observed by light microscopy (×20 magnification). (**C1**) Control, (**C2**) EEV-*GFP* injected, and (**C3**) *GFP*-mRNA injected. Scales bars: 1000 µm. (**D**) Seminiferous tubule structures are not affected by in vivo injection and electroporation with EEV-*GFP* and *GFP*-mRNA. Enlargement of cross sections showing the fine structure of a seminiferous tubule for control (**D1**), EEV-*GFP* (**D2**), and *GFP*-mRNA (**D3**). In each tubule, the different layers of spermatogenic cells are indicated, Sertoli cells (S), spermatogonia (Sg), spermatocytes (Scytes), round spermatids (Stids), mature spermatid cells (m-Sptids), Leydig cells (**L**). Scales bars: 20 µm. (**E**) The area of seminiferous tubules is not affected by in vivo injection and electroporation with EEV-*GFP* and *GFP*-mRNA. Comparison of the seminiferous tubule diameter after injection of nucleic acid vectors or control solutions. Data are represented as a box plot median. The areas of seminiferous tubules (µm$^2$) were measured for round cross sections of n > 35 tubules per testis section (n = 5 testis sections per condition). Statistical significance was verified using a Student's *t*-test. (**F**) Injection/electroporation does not impact epididymal sperm cells. Representative sperm observed by light microscopy on day 7 after injection/electroporation with Control solution (**F1**), EEV-*GFP* (**F2**), or *GFP*-mRNA (**F3**). Scale bars: 10 µm. (**F4**) Percentage of normal epididymal sperm cells in each condition. The number of males was n = 5 for EEV-*GFP*; n = 6 for *GFP*-mRNA, and n = 9 for WT. More than 150 sperm by males were analyzed. Statistical significance was verified using a one-way ANOVA test.

over electroporation. With the protocol we have developed, no such lesions were observed. Next, the testes were measured and weighed. No statistical differences in length and weight were observed between control and treated testes (*Figure 2A2, A3, B2, B3*). Then, microscopic differences were sought by histological analysis of 5 µm sections (*Figure 2C*). No difference was observed between the control condition and EEV-*GFP* or *GFP*-mRNA on the full cross sections (*Figure 2C1–C3*). Next, we observed all the different testicular cells, including all germ cell types and Sertoli cells (*Figure 2D1–D3*) for each condition. The layered structure of germ cells was identical in all conditions. Analysis of the histological sections revealed no differences in the tubules area of the testes injected either with EEV-*GFP* or *GFP*-mRNA (*Figure 2E*). At last, Harris–Shorr staining of the epididymal sperm cells demonstrated that there were no increases in morphological defects when mRNA and EEV were used in comparison with the control (*Figure 2F4*). Taken together, these results suggest that in vivo micro-injection and electroporation of EEV or mRNA did not perturb spermatogenesis.

## Analysis of EEV-*GFP* and *GFP*-mRNA testicular expression by whole testis imaging

After validating the injection method, we compared the kinetics of GFP expression and the maintenance of the fluorescent signals for mRNA and EEV. To do so, we injected and electroporated one testis of adult B6D2 mice with EEV-*GFP* (n = 129) or with *GFP*-mRNA (n = 65). At 0-, 1-, 7-, 15-, 21-, 28-, 35-, 42-, and 49-day post-injection, the whole testes were observed under an inverted microscope. The exogenous fluorescence was directly visible at the surface of the testes when illuminated with light at the appropriate wavelength (*Figures 3 and 4*). No testicular lesions were observed on the testes at any post-injection time (*Figures 3A1–H1 and 4A1–F1*). In addition, both *GFP*-mRNA and EEV-*GFP* induced GFP expression in the testes (*Figures 3A2–H2 and 4A2–F2*). It is worth noting that both vectors induced GFP expression at one day post-injection. However, the duration of fluorescent signals was different. For EEV, GFP fluorescence was still observable on day 42 for 100% of samples, and 56% of samples were positive on day 49, indicating that expression lasted around 1.5 months (*Figure 4G*). In contrast, for mRNA, 100% of testes were labeled on day 21, but none showed any fluorescence on day 28 (*Figure 4G*). Thus, EEV transfection allowed a considerably longer duration of expression than mRNA (*Figures 3 and 4*). It is important to underline that the signal measured is the fluorescence emitted by the GFP. This signal is dependent on both the half-lives of the plasmid/mRNA and the GFP. Therefore, the kinetic of the signal persistence (which is called here expression) is a combination of the persistence of the vector and the synthesized protein. In addition to differences in duration of expression, the GFP expression patterns were clearly different: mRNA produced a large, diffuse pattern, highlighting the shape of the seminiferous tubules; EEV-*GFP* produced a punctiform pattern (*Figures 3B and 4B*).

These results suggest that *GFP*-mRNA and EEV-*GFP* targeted different seminiferous cell types, such as Sertoli cells and all germline cells, or that there were differences in terms of transfection efficiency. Moreover, the stability of mRNA-GFP was assessed by RT-qPCR in HEK cells and seminiferous tubule cells (*Figure 5*). mRNA-GFP was detected for up to 60 hr in HEK cells and for up to two weeks in seminiferous tubule cells (*Figure 5*). Together, these results suggest that the long-lasting

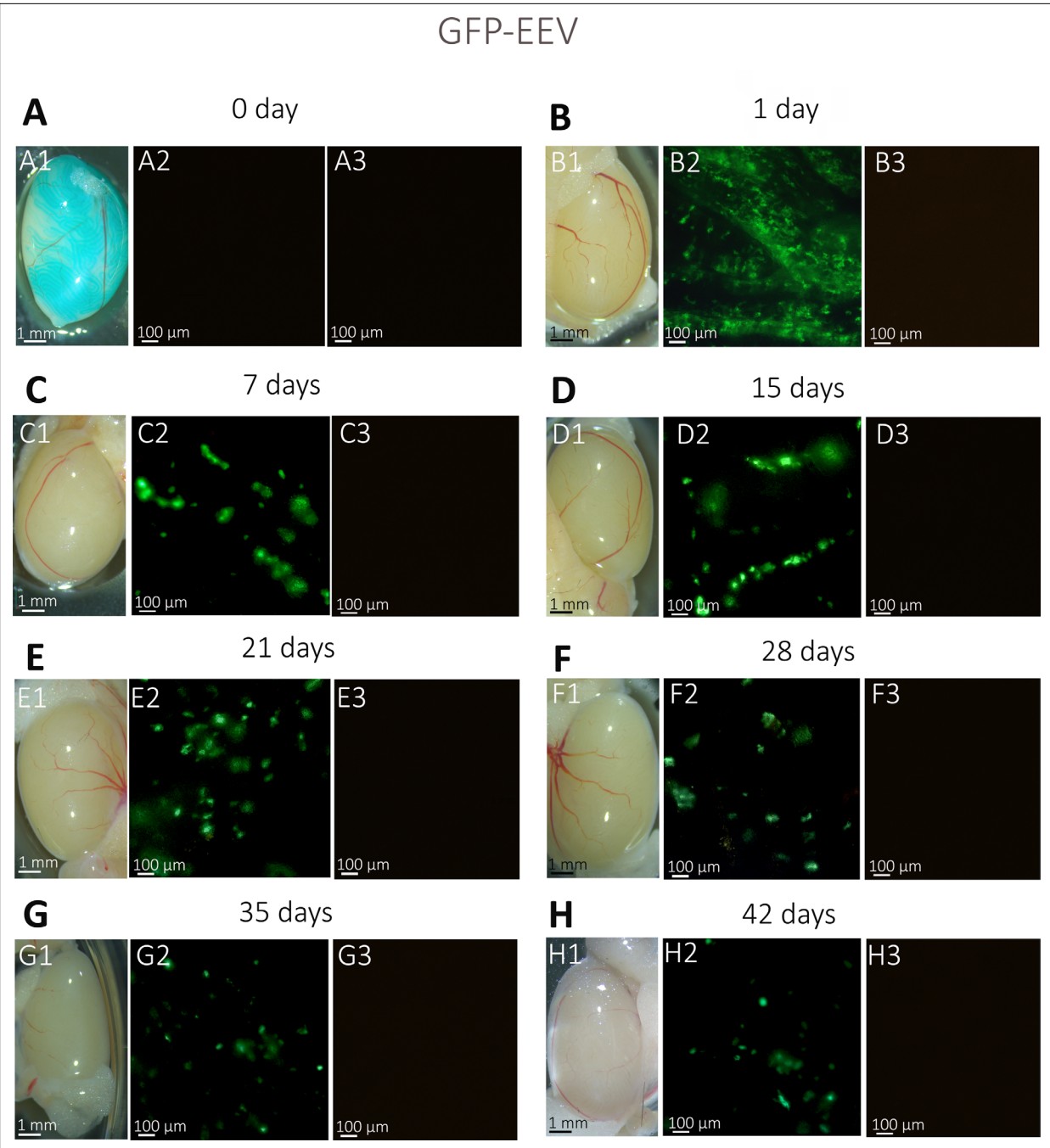

**Figure 3.** Kinetics of EEV-*GFP* expression following in vivo injection/electroporation: whole testicular expression. (**A1–H1**) Whole-mount testes on days 0, 1, 7, 14, 21, 28, 35, and 42 after in vivo injection/electroporation with EEV-*GFP*. (**A2–H2**) Under fluorescence observation, GFP expression was detectable in transfected testes from 12-week-old B6D2 mice. (**C3–H3**) Insets show the absence of autofluorescence in non-transfected control testes, observed under ×4 magnification. The GFP expression presented a punctiform pattern in seminiferous tubules and was detected from 1 to 42 days. Scales bars: 1 mm and 100 μm.

fluorescence observed in our experiments reflects a combination of transcript stability, local translation within germ cells, and the slow protein turnover that is typical of the spermatogenic lineage.

## Kinetics of EEV and mRNA expression assessed by in vivo imaging

To further assess and compare the kinetics of the expression of the two vectors, we expressed exogenous luciferase in the testis using EEV or mRNA and observed the level of luciferase by in vivo

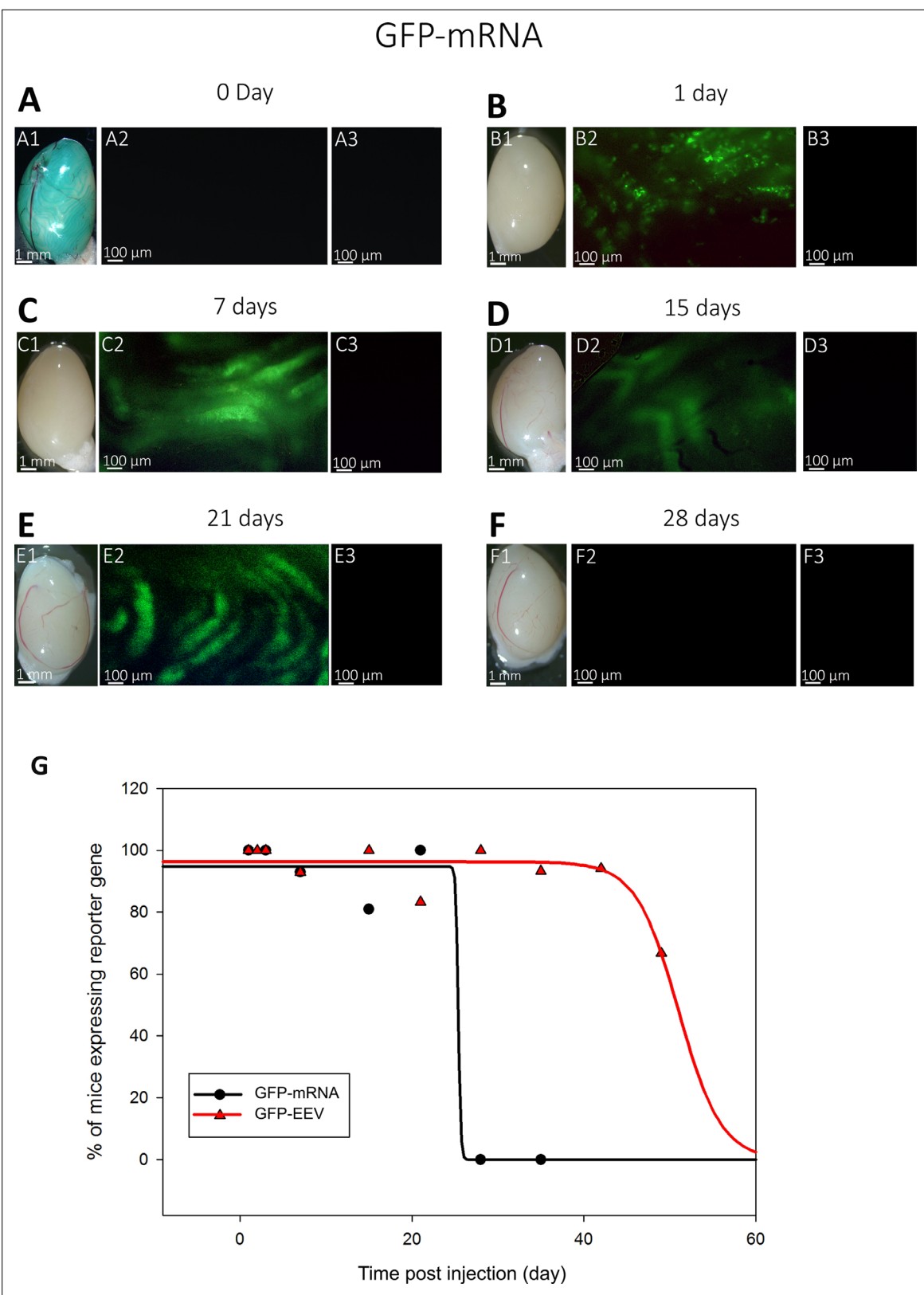

**Figure 4.** Kinetics of *GFP*-mRNA expression following in vivo injection/electroporation: whole testicular expression. (**A1–F1**) Whole-mount testes on days 0, 1, 7, 15, 21, and 28 after in vivo injection/electroporation with *GFP*-mRNA. (**A2–F2**) Under fluorescence observation, GFP expression was detectable in transfected testes from 12-week-old B6D2 mice. (**A3–F3**) Insets show the absence of autofluorescence in non-transfected control testes, observed under ×4 magnification. The GFP expression presented a continuous pattern in seminiferous tubules and was detected from day 1 to day 15.

*Figure 4 continued on next page*

*Figure 4 continued*

Scale bars: 1 mm and 100 µm. (**G**) Comparison of the percentage of injected mice exhibiting reporter gene expression. Mice injected with *GFP*-mRNA exhibited GFP expression from day 1 to day 21. Mice injected with EEV-GFP exhibited GFP expression from day 1 to day 49 (for EEV-*GFP* $n$ = 11 on day 1; $n$ = 13 on day 2; $n$ = 10 on day 3; $n$ = 14 on day 7; $n$ = 5 on day 10; $n$ = 12 on day 15; $n$ = 11 on day 21; $n$ = 12 on day 28; $n$ = 15 on day 35; $n$ = 17 on day 42 and $n$ = 9 on day 49; for *GFP*-mRNA $n$ = 3 on day 1; $n$ = 4 on day 3; $n$ = 15 on day 7; $n$ = 21 on day 15; $n$ = 15 on day 21, and $n$ = 5 on day 28).

The online version of this article includes the following figure supplement(s) for figure 4:

**Figure supplement 1.** Testicular expression of *mCherry*-mRNA following in vivo electroporation.

**Figure supplement 2.** Decay over time of the number of mice exhibiting reporter gene expression following injection/electroporation of the three different mRNAs.

bioluminescence imaging. For EEV, we took advantage of the fact that the EEV plasmid contains the DNA sequence of the luciferase protein (*CAGs-GFP-T2A-Luciferase*) in addition to the DNA sequence of the GFP fluorescent protein (*Figure 1—figure supplement 1*). For mRNA, we injected commercial naked *luciferase*-mRNA into the *rete testis* according to the same protocol as used for *GFP*-mRNA. For this set of experiments, we injected the EEV-*GFP-Luc* and *luciferase*-mRNA into the testes of 6 adult mice on day 0. We injected a similar number of copies of mRNA and EEV. The testes were imaged in vivo to detect bioluminescence expression, on day 1 and weekly until disappearance of the signal, no more than 120 days post-injection (*Figure 6*). For EEV-*GFP-Luciferase*, the bioluminescence induced by transfection was detected from day 1. After a rapid decrease in signal intensity over the first 3 weeks, a weak but constant signal remained detectable for 3 months, then faded away (*Figure 6A1, A2*). For *Luciferase*-mRNA, expression was also detected from day 1. The bioluminescence decreased gradually over 3 weeks, becoming undetectable thereafter (*Figure 6B1, B2*). These results are consistent with our previous results (*Figure 3*) and confirm that EEV allows a longer expression of exogenous protein within the testis. *Figure 6C* compares the kinetics of expression observed with EEV and mRNA. Overall, our results indicated a stronger expression for mRNA than for EEV, but with expression decreasing rapidly over time, and almost no remaining signal after 3 weeks. In contrast, residual expression was detectable for several months with EEV.

## Assessing testicular and cellular *GFP*-mRNA expression using whole testicle optical clearing, lightsheet microscopy, and 3D reconstructions

To better characterize the spatial distribution of *GFP*-mRNA expression in the testis, we performed whole testicular optical clearing. On day 0, we injected and electroporated testes with *GFP*-mRNA ($n$ = 6). On day 1, we harvested the testes and cleared them with a modified optical clearing protocol, as described in the MM section. After complete clearing (*Figure 7A*), we imaged the whole testes using a lightsheet microscope and performed 3D reconstruction from the images stack obtained (*Videos 1 and 2* and *Figure 7B*). From this 3D reconstruction, we determined the volume of the testis stained with GFP by measuring the GFP-positive area in each image and multiplying it by the thickness of the z-step (10 µm). Due to optical issues, only half of the whole testis was acquired, then the sample was turned by 180° to acquire the other half. A total GFP-stained volume of 0.51 and 0.23 mm$^3$ was determined from face A and face B, respectively. The corresponding total volume of half part of the testis was measured as 60 mm$^3$; therefore, 0.81% and 0.24% of the testis was transfected for face A and face B, respectively (*Figure 7B*).

## Assessing GFP cellular expression using whole testicle optical clearing and adaptive optics confocal microscopy

Because the GFP fluorescence patterns were different for the two nucleic vectors when observed under the inverted microscope (*Figure 3*), we wondered whether the same cell types were targeted in both cases. To address this question, the whole optical cleared testes from EEV-*GFP* and *GFP*-mRNA-transfected mice were imaged using a confocal microscope. The different cell types were identified based on their positions within the seminiferous tubule, their cellular shape, and their nuclear morphology - revealed by nuclear staining. For instance, Sertoli cells have an oval to elongated nucleus and the cytoplasm presents a complex shape ('tombstone' pattern) along the basement membrane, with long projections that extend toward the lumen (*Ruthig and Lamb, 2022*). Round spermatids have small, round, and compact nuclei with a nucleolus and are localized between the spermatocytes

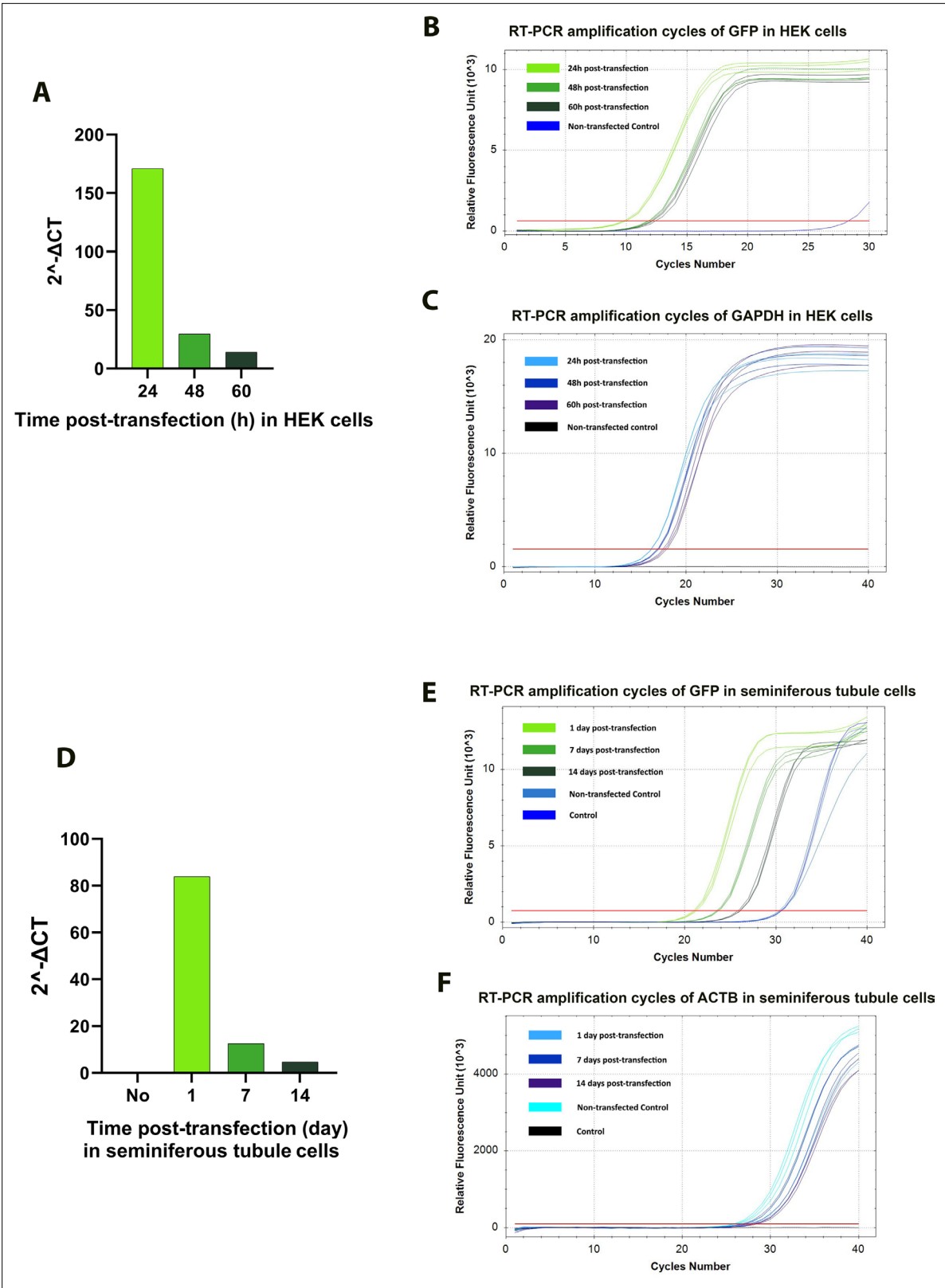

**Figure 5.** Stability of mRNA-GFP in HEK cells and seminiferous tubule cells. (**A–C**) Analysis of mRNA-GFP stability in HEK cells. (**A**) Quantification of relative GFP mRNA levels ($2^{-\Delta CT}$) at 24-, 48-, and 60-hr post-transfection. (**B**) RT-qPCR amplification curves of GFP and (**C**) GAPDH transcripts at the indicated time points, showing a progressive decrease in GFP signal over time while GAPDH expression remained constant. (**D–F**) Analysis of mRNA-GFP stability in mouse seminiferous tubule cells after in vivo injection and electroporation. (**D**) Quantification of relative GFP mRNA levels ($2^{-\Delta CT}$) at 0-, 1-,

*Figure 5 continued on next page*

*Figure 5 continued*

7-, and 14-day post-transfection. (**E**) RT-qPCR amplification curves of GFP and (**F**) ACTB transcripts at corresponding time points. mRNA-GFP remained detectable for up to two weeks in seminiferous tubules, indicating enhanced transcript stability in the testicular environment.

and elongated spermatids (*de Boer et al., 2015*). Depending on the stage of the seminiferous epithelium, round spermatids (steps 1–8) and elongated spermatids (steps 9–16) can be observed in adjacent regions of the same tubule, reflecting the continuous and overlapping progression of spermatid differentiation rather than their constant coexistence within the same stage. For EEV-*GFP*, on day 1 post-injection and electroporation, a strong punctiform green fluorescence was visible inside the seminiferous tubules (*Figure 8A*). Based on the different morphological criteria, this fluorescent signal was detected in spermatocytes, round spermatids, and Sertoli cells (*Figure 8B, C*). On day 7, the GFP signal induced by EEV-*GFP* was reduced and only isolated signals in a few seminiferous tubules (1 per 11 tubules) were observed (*Figure 8D*). These signals were associated only with Sertoli cells (*Figure 8E*).

Germ cells are non-adherent in culture, which allowed us to distinguish them from potential somatic cells that may not have been completely lysed during the dissociation step. Moreover, germ cells have specific sizes and nuclear morphologies. For example, round spermatids are small and possess small, round, compact nuclei with a nucleolus. Nuclear staining with Hoechst allows us to examine the nuclear morphology of cells expressing GFP and, by combining this observation with cell size measurements, to determine the cell type.

Thus, 24-hr post-injection and electroporation, we were able to observe that GFP was expressed in spermatocytes, round spermatids, and elongated spermatids.

For the mRNA vector, on day 1 and 7 post-injection and electroporation, an intense fluorescence was observed in all the germ cells and in Sertoli cells in the seminiferous tubules (*Figure 9A*). At the cellular level, this fluorescent signal was associated with spermatogonia, spermatocytes, round spermatids, elongated spermatids, and mature spermatids cells to similar extents for both post-injection times (*Figure 9B, D*). In contrast to what was observed with EEV on day 7, no reduction in the number of fluorescent seminiferous tubules was noted when using *GFP*-mRNA, with 8 out of 10 tubules stained (*Figure 9C, D*).

## Assessing GFP-mRNA expression in dissociated germ cells following in vivo testicular electroporation

To assess the efficiency of in vivo mRNA delivery into testicular germ cells, GFP-mRNA was injected and electroporated into the testes on day 0. On day 1, testes were collected, enzymatically dissociated (Fig supp 10), and the resulting seminiferous tubule cell suspensions were cultured for 12 hr. Live cells were then analyzed by fluorescence microscopy (*Figure 10*). We observed GFP expression in various germ cell types, including pachytene spermatocytes (53.4%) (*Figure 10A*), round spermatids (25%) (*Figure 10B–E*) and in elongated spermatids (11.4%) (*Figure 10C–E*). Fluorescence imaging revealed strong cytoplasmic GFP signals in each of these populations, confirming efficient transfection and translation of the delivered mRNA. These results demonstrate that the in vivo injection and electroporation protocol enables effective mRNA transfection across multiple stages of spermatogenesis.

## Expression of naked *mCherry*-mRNA in testis following electroporation

Heterologous expression is well known to depend on the protein studied; we therefore tested the same reporter proteins to compare the mRNA and EEV vectors in the experiments presented above. Apart from the bioluminescence experiments with luciferase, we compared only GFP protein expression. To validate and confirm the capacity of naked mRNA to express proteins in the testes after injection and electroporation, we further challenged the method with mCherry, another reporter protein (*Figure 1—figure supplement 1B–D*). We injected homemade naked mRNA coding for mCherry into the testes. As previously with *GFP*-mRNA, no testicular lesions were observed (*Figure 4—figure supplement 1A1, B1, C1,D1, E1, F1*).

The assessment of the temporal persistence of testicular mCherry fluorescent protein expression revealed a robust red fluorescence from day 1 post-injection, which remained detectable for at least 15 days (*Figure 4—figure supplement 1B2, C2, D2*). At the cellular level, the fluorescent signal was

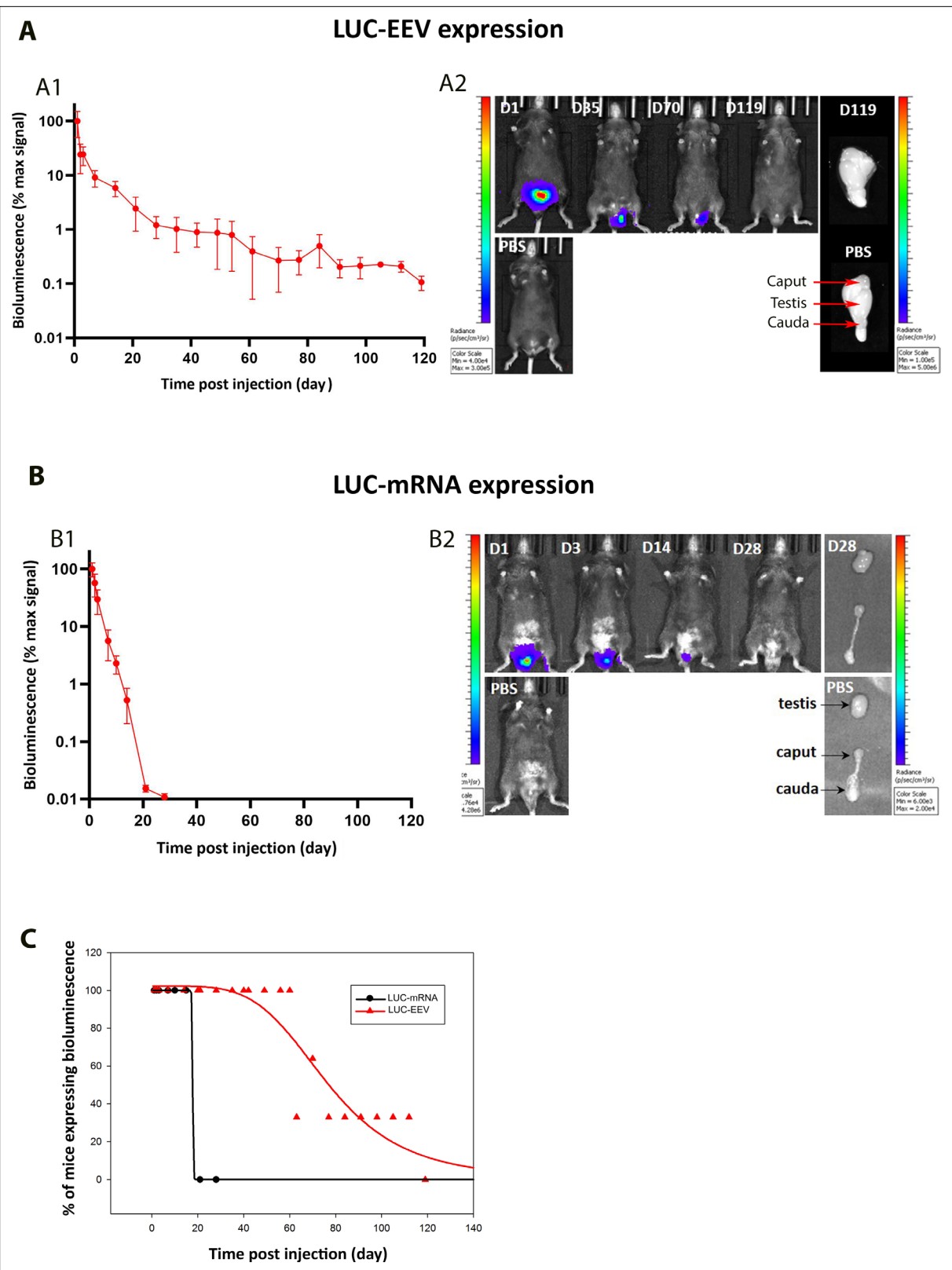

**Figure 6.** Kinetics of EEV and mRNA expression by in vivo bioluminescence imaging. (**A**) In vivo bioluminescence imaging quantification of luciferase expression over time following injection/electroporation of EEV-*GFP-luc*. (**A1**) EEV-*GFP-Luc* was injected into the testes and electroporated on day 0. Bioluminescence signal was quantified at several time points. Results are expressed as a percentage of the maximal signal (mean ± SEM; *n* = 5 mice up to D2; *n* = 4 from D3 to D28; *n* = 3 from D35 to D98; and *n* = 3 from D105 to D119). (**A2**) In vivo bioluminescence images of a representative mouse at

*Figure 6 continued on next page*

*Figure 6 continued*

several time points after administering EEV-*GFP-LUC* or PBS, and ex vivo bioluminescence images of testes after 119 days. (**B**) In vivo bioluminescence imaging quantification of luciferase expression over time induced by injection/electroporation of *LUC*-mRNA. (**B1**) *LUC*-mRNA was injected into the testes and electroporated on day 0. Bioluminescence signal was quantified in the whole testis at several time points. Results are expressed as a percentage of the maximal signal (mean ± SEM; *n* = 5 mice). (**B2**) In vivo bioluminescence images of a representative mouse at several time points after administering *LUC*-mRNA or PBS, and ex vivo bioluminescence images of caput, testes, and cauda after 28 days. (**C**) Decay over time of the number of mice expressing reporter genes. Mice were injected on day 0 with *LUC*-mRNA or EEV-*GFP-LUC* and the number of mice showing bioluminescence in the testis was counted at different time points, from day 1 to day 119. For EEV-*GFP*: *n* = 13 at D1; *n* = 13 at D2; *n* = 4 from D3 to D28; *n* = 3 from D35 to D98; and *n* = 3 from D105 to D119. For *LUC*-mRNA: *n* = 5 mice for all-time points.

detected in germ cells, including spermatogonia, spermatocytes, round spermatids, mature spermatids, and Sertoli cells on days 1 and 7 post-injection (*Figure 9—figure supplement 1*).

Finally, we compared the kinetics and levels of expression of the three different mRNA molecules, coding for mCherry, GFP, and luciferase. By comparing the number of mice expressing *mCherry*-mRNA, *GFP*-mRNA, and *luciferase*-mRNA fluorescence/luminescence over 21 days, we observed first that expression was detectable for all mRNAs on day 1, and second that the duration of expression was slightly different for the individual mRNAs. For instance, on day 15, 100%, 80%, and 60% of mice injected with *GFP*-mRNA, *mCherry*-mRNA, and mRNA-*luciferase*, respectively, presented fluorescence/bioluminescence, and on day 21, 100% of mice expressed GFP, whereas no signal was observed for mCherry or Luciferase (*Figure 4—figure supplement 2*).

## Endogenous expression of ARMC2 in germ cells

Before attempting to rescue expression, we felt it was important to better characterize *Armc2* expression in healthy germ cells, and in particular to study the timing of expression.

IF was used to determine when ARMC2 protein was detectable during spermatogenesis. For these experiments, dissociated cells from testes were observed to detect the presence of ARMC2 on different spermatogenic cells. ARMC2 was present only in the flagella of the elongated spermatids (*Figure 11A*, *Figure 11—figure supplement 1A*). The specificity of the signal was validated using testicular cells from Armc2 KO mice, where no signal was observed on all spermatogenic cells (*Figure 11B*). In transversal sections of WT seminiferous tubules, ARMC2 signal was not present in spermatogonia and spermatocytes (*Figure 11—figure supplement 1B*), but detected in spermatid layers.

By analyzing the RNA-seq database produced by Gan's team (*Gan et al., 2013*), we show that the mRNA encoding ARMC2 starts to be expressed at the pachytene spermatocyte stage, then shows a gradual increase at the round spermatid stage, and finally becomes predominantly expressed at the elongated spermatid stage (*Figure 11—figure supplement 1C*), a result in agreement with a post-meiotic function of the protein. Finally, we also consistently observed a staining at the base of the manchette of elongating spermatids, but we have no explanation for that (*Figure 11A3*).

In conclusion, the results presented here demonstrate that the ARMC2 protein is expressed and translated at the late stages of spermatogenesis.

## Co-injection of *Armc2*-mRNA and *eGFP*-mRNA followed by electroporation is safe and induces green fluorescence in the seminiferous tubules

We next tested whether the injection and electroporation of *Armc2*-mRNA molecules had any deleterious effects on testis anatomy and seminiferous tubule structure. We first verified the quality of *Armc2*-mRNA synthesis by transfecting HEK cells and performing Western blot (*Figure 12—figure supplement 1*). After this validation, we co-injected *Armc2*-mRNA and *eGFP*-mRNA into the left testes of mice, using the right testes as untreated controls. *eGFP*-mRNA was co-injected to verify and monitor transfection efficiency. The testes were observed under a binocular microscope at different times (3, 6, 10, 15, 21, 28, and 35 days) after electroporation to identify possible macroscopic degeneration of the seminiferous tubules. No morphological defects were observed in the testes co-injected with *Armc2*-mRNA and *eGFP*-mRNA. An example of control and injected testes from day 15 is presented in *Figure 12A1, B1*. The testes were also weighed at different times post-injection, and the weight ratios of injected testes to non-injected control testes were determined. For all-time points,

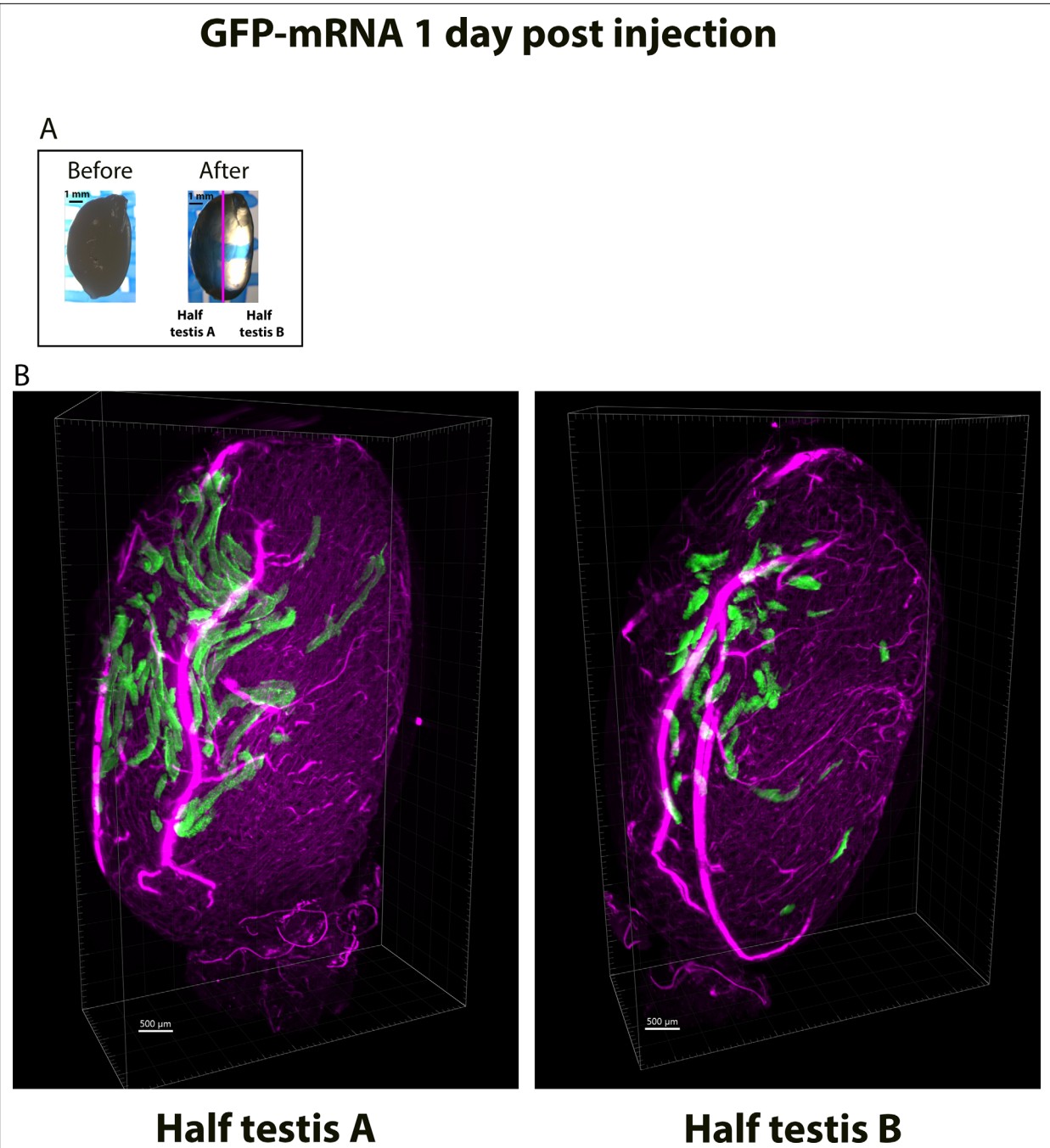

**Figure 7.** Testicular and cellular *GFP*-mRNA expression measured on optically cleared testis after 3D image reconstructions from lightsheet microscopy imaging. Testes were injected/electroporated with *GFP*-mRNA on day 0. On day 1, whole testes were fixed and subjected to optical clearing. (**A**) Testes were observed before and after optical clearing on a binocular microscope. The right image shows the transparency of the testis after complete clearing, revealing the blue mesh throughout the organ. (**B**) The 3D internal structure of a cleared testis was reconstructed from the lightsheet microscopy images. The reconstruction was possible only for a half testis due to optical issues. Two opposing faces of the same testis are presented, allowing the distribution of GFP fluorescence throughout the seminiferous tubules to be measured. Pink fluorescence corresponds to the autofluorescence of interstitial cells located around the seminiferous tubules. Scale bars A: 1 mm and B: 500 µm.

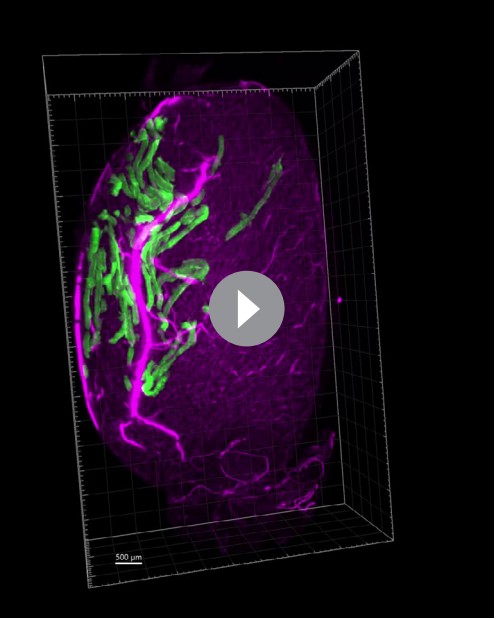

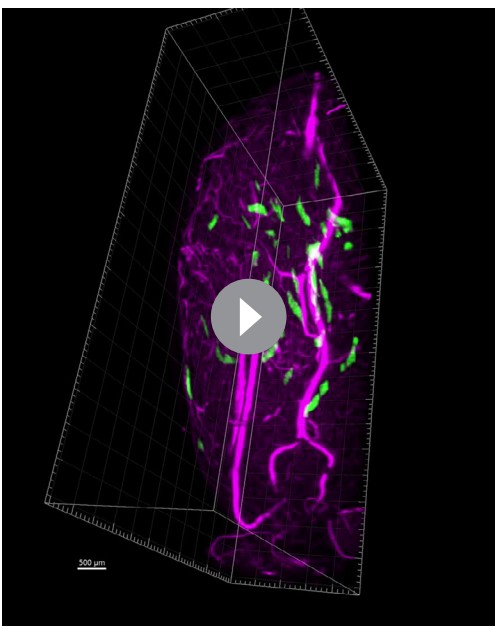

**Video 1.** 3D-microscopic reconstructions of faces 1 and 2 of a testis injected with *GFP*-mRNA.
https://elifesciences.org/articles/94514/figures#video1

**Video 2.** 3D-microscopic reconstructions of faces 1 and 2 of a testis injected with *GFP*-mRNA.
https://elifesciences.org/articles/94514/figures#video2

this ratio was close to 1 (*Figure 12C*), confirming that the method and the mRNAs did not cause any injury at the organ level. Next, under blue light, the efficiency of the transfection was assessed by observing the GFP fluorescence at the surface of the testes. GFP fluorescence was observed on testes injected with *Armc2*-mRNA and *eGFP*-mRNA 2 weeks after injection (*Figure 12B2*), indicating that the naked mRNA was successfully transfected into testicular cells.

## Motile sperm cells detected in *Armc2* KO mice following *Armc2*-mRNA injection and electroporation into testes

We then assessed whether the injection of *Armc2*-mRNA into the testes in *Armc2* KO mice restored sperm motility. We examined the motility of sperm cells present in the caudal part of the epididymis at different times post-injection (3- to 35-day post-injection). For each condition, between 3 and 16 KO mice were used.

The *Armc2* KO model used is known to produce sperm cells with short and irregular flagella that are therefore immotile on day 0. No motile sperm were observed on days 3, 6, 10, 15, or 28 after surgery (*Figure 13A*). However, motile sperm cells were found in the epididymis of some *Armc2* KO mice at 21 and 35 days post-treatment (*Figure 13A*). Indeed, one in three mice had motile sperm at 21 days post-surgery, rising to three in four mice at 35 days post-injection. Nevertheless, the number of motile sperm observed remained low: 5.5% after 21 days and 7.15% after 35 days post-injection (*Figure 13A1*). Although the number of motile sperm remained low: 5.5% at 21 days and 7.15% at 35 days post-injection (*Figure 13A1*), this recovery represents a substantial improvement considering the small fraction of germ cells reached by the current electroporation method (*Figure 6B*). The sperm motility parameters of *Armc2*−/−-rescued motile sperm were characterized in comparison to those of *Armc2*+/+ sperm using the CASA system (*Figure 13A2*). These parameters included VAP, VSL, VCL, ALH, BCF, and STR. We have observed significant differences between WT and rescued sperm. In particular, the VSL and LIN parameters are lower for rescued sperm. Next, sperm were sorted as progressive, intermediate, hyperactivated, or slow according to motility parameters of motile sperm and recorded from their track (*Figure 13A3*). The percentage of hyperactivated sperm and the proportion of intermediates in the *Armc2*-/--rescued motile sperm population were found to

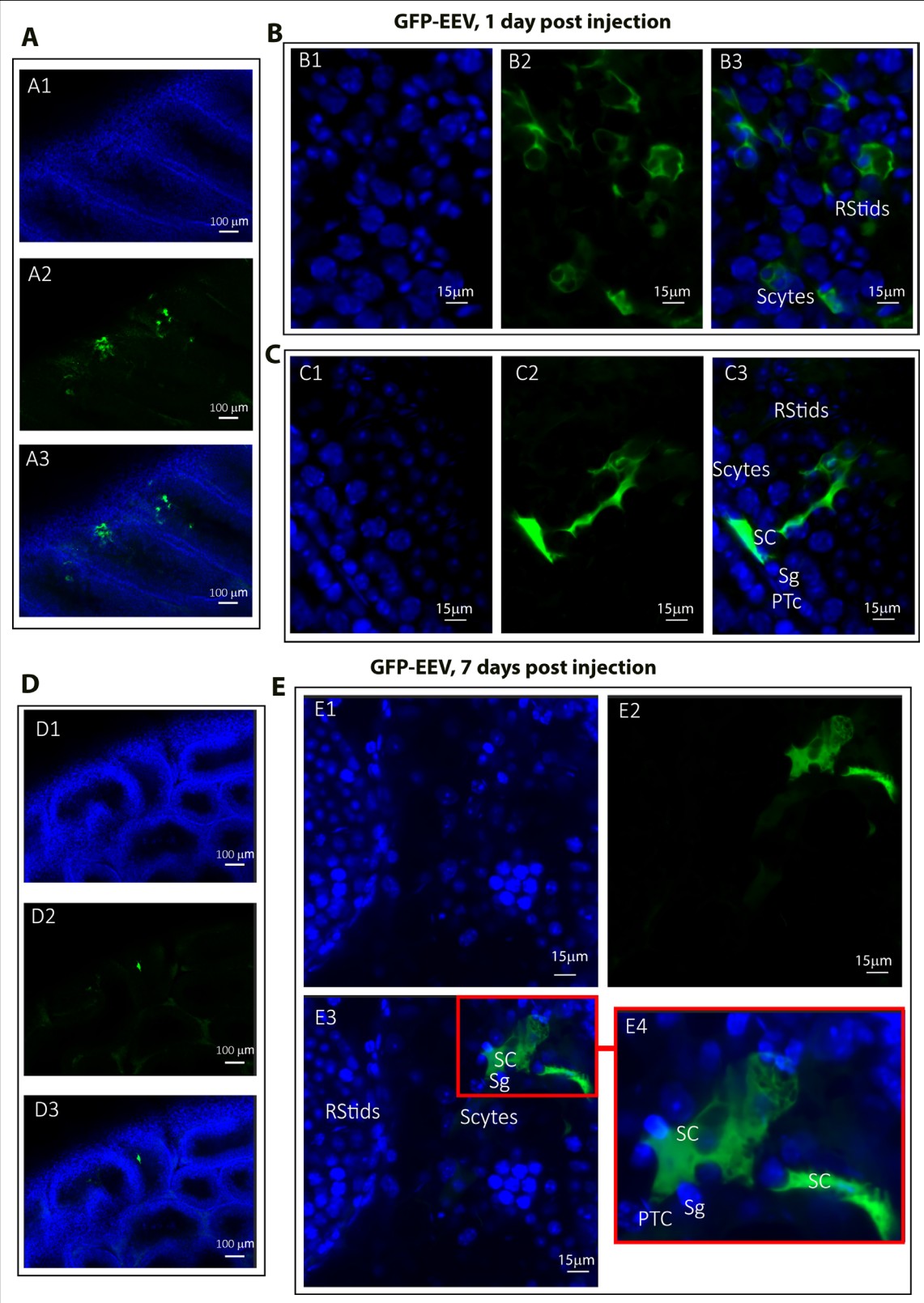

**Figure 8.** Cellular expression of EEV-*GFP* following in vivo injection/electroporation. Testes were injected/electroporated with EEV-*GFP* on day 0. On day 1 and on day 7, whole testes were fixed and subjected to optical clearing. Cleared tests were observed by fluorescence microscopy. (**A1–A3**) On day 1, transfected seminiferous tubules showed dotted green fluorescence at low magnification (×10/0.45). Nuclei were counterstained with DAPI (blue staining) to reveal the structure of the seminiferous tubules. At the cellular level, fluorescence was detectable (**B1–B3**) in germ cells including

*Figure 8 continued on next page*

*Figure 8 continued*

spermatogonia (Sg), spermatocytes (Scytes), and round spermatids (RStids), as well as (**C1–C3**) in Sertoli cells (SC). (**D1–D3**) On day 7, the GFP signal was lower at low magnification (×10/0.45) and detectable (**E1–E3**) only in Sertoli cells (×40/1.15 WI) (*n* = 3) (PTc = peri-tubular myoid cell). **E4** is an enlargement of the red square in E3, allowing the cell type to be identified. Scale bars: 100, 15, and 3 µm.

be increased in comparison to the control. Videos showing sperm motility in different conditions are available in the online material associated with this article (*Videos 3–6*).

After verifying motility, we looked at the morphology of the spermatozoa present in the cauda epididymis. Six days after injection of *Armc2*-mRNA, the cells detected were mostly round cells and abnormal spermatozoa with a short or coiled flagellum measuring between 7 and 20 µm. The same cell types were observed at 3-, 10-, 15-, and 28-day post-surgery. In contrast, the motile sperm detected on days 21 and 35 had a normal morphology with a long flagellum (greater than 100 µm) and a hook-shaped head (*Figure 13*, *Figure 13—figure supplement 1*).

## Motile sperm cells detected in *Armc2* KO mice following *Armc2*-mRNA injection and electroporation into testes are fertile

We subsequently evaluated the efficacy of *Armc2*-mRNA injection into the testes of *Armc2* KO mice in restoring sperm fertility. The fertility outcome was assessed through in vitro fertilization (IVF) and ICSI experiments. The sperm rescued from *Armc2*$^{-/-}$ mice were capable of successfully fertilizing eggs and producing embryos at the 2-cell stage by IVF (*Figure 14A–C*). The 2-cell embryo formation observed after IVF serves as a strong indication of partial functional recovery. Notably, 62.7% of 2-cell embryos were obtained with the *Armc2*$^{-/-}$-rescued sperm by IVF, compared to only 2.67% with the *Armc2*$^{-/-}$ sperm. Three percent of eggs became 2-cell embryos when fertilized with sperm from *Armc2*$^{-/-}$, a rate not significantly different to that observed for eggs incubated 24 hr without sperm, and likely corresponding to parthenogenesis activation (*Figure 14A1–A4, B*).

To gain further insight, a comparative analysis of the developmental outcomes of mouse embryos generated by ICSI with spermatozoa from wild-type (WT), *Armc2*$^{-/-}$, and *Armc2*$^{-/-}$ treated mice was performed. It should be noted that in the case of *Armc2*$^{-/-}$-treated mice, the ICSI procedure was performed only with the motile *Armc2*$^{-/-}$-rescue spermatozoa. The percentage of live injected oocytes that reached the blastocyst embryos was 46% for WT spermatozoa, 25% for *Armc2*$^{-/-}$-rescued spermatozoa, and 13% for *Armc2*$^{-/-}$ spermatozoa (*Figure 9—figure supplement 1*) .The findings indicate that the developmental potential of the embryos was enhanced when *Armc2*$^{-/-}$-rescued sperm were utilized as opposed to *Armc2*$^{-/-}$ sperm.

Overall, these results demonstrate that the *Armc2*$^{-/-}$-rescue motile spermatozoa can successfully fertilize eggs.

## Discussion

The challenge of treating male infertility remains to be addressed. Current assisted reproduction techniques are unable to treat all patients, and alternative strategies need to be developed to meet the legitimate desire to be a father. The aim of this study was to evaluate the potential of naked mRNA as a means to induce an expression of exogenous proteins in male germ cells in a preclinical adult mouse model. Based on previous studies using electroporation, we investigated whether the combination of the injection of naked mRNA and in vivo electroporation could lead to an efficient protein expression in spermatogenic cells. We chose to first study the efficiency of capped and poly(A)-tailed mRNA coding for reporter proteins and compared the results to those obtained with a non-integrative EEV plasmid. No EEV plasmid has ever been tested in the context of infertility treatment before this study.

Using an adult mouse model, we optimized the micro-injection and electroporation method described by *Michaelis et al., 2014*. We show that the microinjection through the *rete testis* did not provide a homogenous distribution of the particles throughout the seminiferous tubules. Nevertheless, the seminiferous tubules remained intact, with no signal detected in the peritubular space. The peripheral expression observed was due to the close vicinity of cells to the electrodes and to a peripheral dispersal of the injected solution, as shown by the distribution of the fluorescent i-particles NIRFiP-180. Our results also showed that the combination of injection and electroporation did not perturb spermatogenesis when electric pulses were carefully controlled. Using such a protocol, we

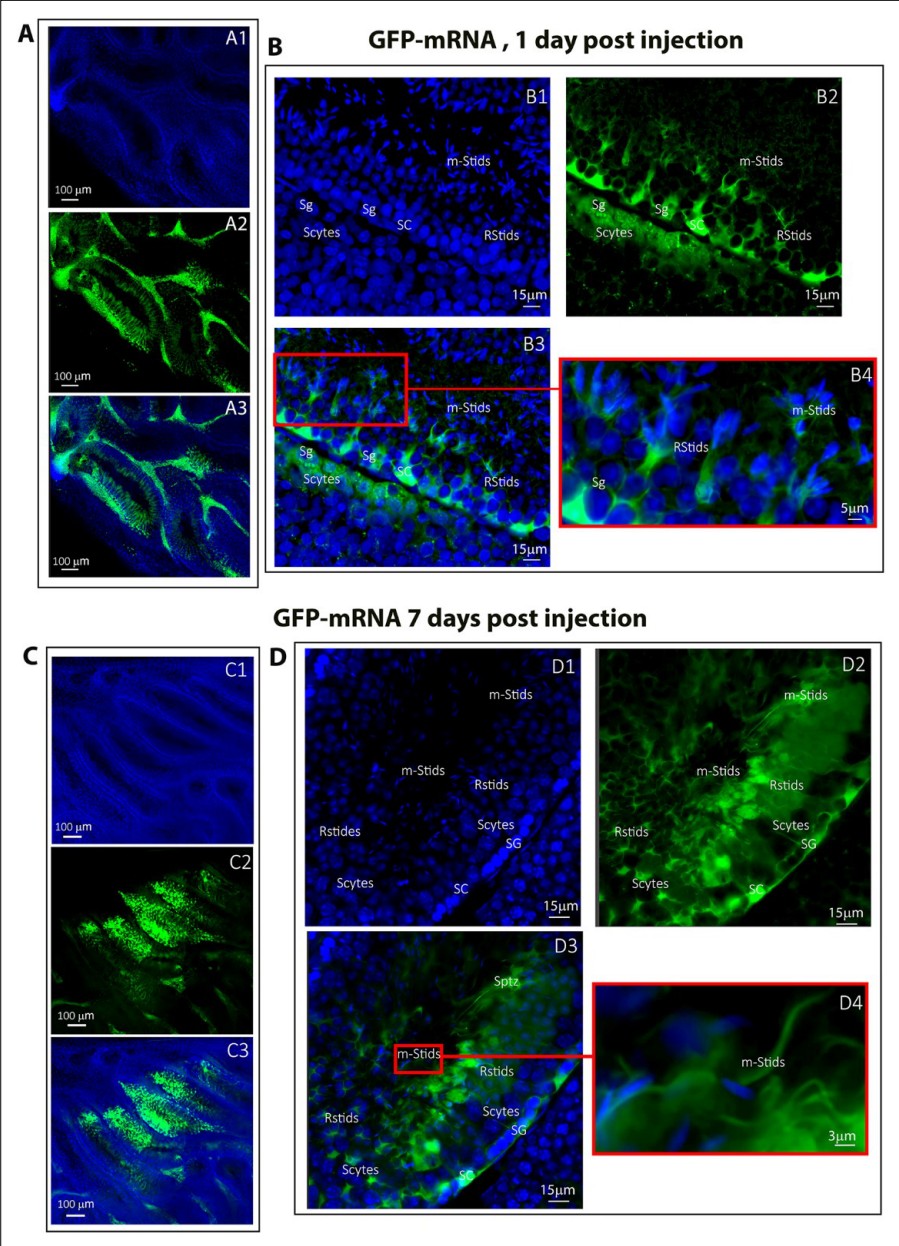

**Figure 9.** Cellular expression of *GFP*-mRNA following in vivo injection/electroporation. Testes were injected/electroporated with *GFP*-mRNA on day 0. On days 1 and 7, whole testes were fixed and subjected to optical clearing. Cleared testes were observed by fluorescence microscopy. (**A1–A3**) On day 1, transfected seminiferous tubules showed strong broad-ranging green fluorescence at low magnification (×100/0.45). Nuclei were counterstained with DAPI (blue staining) to reveal the structure of the seminiferous tubule. At the cellular level, fluorescence was detectable in germ cells (**B1–B3**) including spermatogonia (Sg), spermatocytes (Scytes) and round spermatids (RStids), mature spermatid cells (m-Sptids) and Sertoli cells (SC). B4 is an enlargement of the red square in B3, allowing the cell types to be identified. (**D1–D3**) On day 7, the GFP signal remained strong at low magnification (×10/0.45) and was still detectable in (**E1–E3**) all germ cell types and Sertoli cells (×40/1.15 WI) (*n* = 3). **E4** is an enlargement of the red square in E3, showing that testicular sperm were also stained. Scale bars: 100, 15, and 3 µm.

The online version of this article includes the following figure supplement(s) for figure 9:

**Figure supplement 1.** Cellular expression of *mCherry*-mRNA following in vivo injection/electroporation.

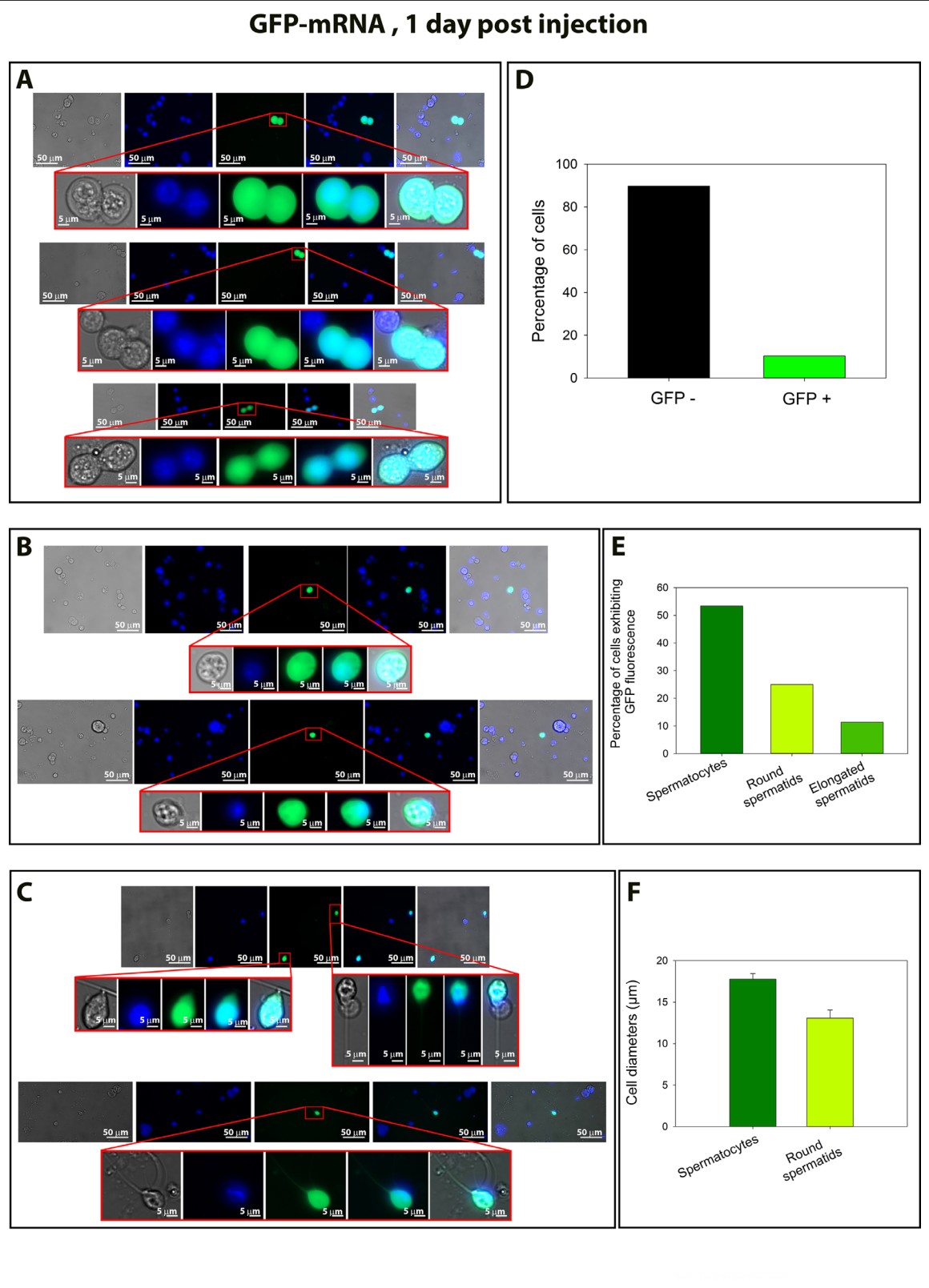

**Figure 10.** GFP-mRNA expression in germ cells following in vivo injection and electroporation. Testes were injected and electroporated with GFP-mRNA on day 0. On day 1, the testes were dissociated, and the resulting seminiferous tubule cell suspension was cultured for 12 hr. Live seminiferous cells were then observed by fluorescence microscopy (LD plan-Neofluar 40x/0.6 Korr M27). (**A**) Representative images of GFP expression in elongated sperm. Panels show brightfield, DAPI staining (nuclei in blue), GFP fluorescence (green), merged channels, and overlay with brightfield. Magnified

*Figure 10 continued on next page*

*Figure 10 continued*

views of selected cells are shown below each row, highlighting strong cytoplasmic GFP signals in elongated sperm. (**B**) Representative images of GFP expression in pachytene spermatocytes. Same panel layout as in (**A**). Enlarged views show paired pachytene spermatocytes displaying robust cytoplasmic GFP fluorescence. (**C**) Representative images of GFP expression in round spermatids. Same panel layout as above, showing isolated round spermatids with clear GFP fluorescence in the cytoplasm. Higher magnification confirms successful mRNA transfection and translation. Scale bars: 50 and 5 μm. (**D**) Percentage of GFP-positive and GFP-negative cells in the analyzed population. (**E**) Distribution of GFP-positive cells across different germ cell types, showing higher uptake in spermatocytes compared with round and elongated spermatids. (**F**) Average diameters of spermatocytes and round spermatids, indicating cell size differences in the population analyzed.

The online version of this article includes the following figure supplement(s) for figure 10:

**Figure supplement 1.** Expression of GFP following mRNA injection in seminiferous tubules.

were able to induce the expression of three reporter proteins, GFP, mCherry, and luciferase in the testis by mRNA injection/electroporation.

Using whole testicular optical clearing, the reporter proteins were synthesized from the injected mRNA in different types of cells including Sertoli cells, spermatogonia, spermatocytes, round spermatids, and mature spermatids from day 1, and were still detectable after 1 week. These results deserve two comments: first, the expression is very fast and synthesized protein is detectable 24 hr injection; second, all cell types have the ability to translate mRNAs. Furthermore, the fact that we observed motile sperm at 21 days after injection confirms that spermatids are transfected and that the translation of the protein of interest is possible at this stage. For EEV, we have a similar result at day 1. However, the yields of seminiferous tubules and cellular transfection are lower. In particular, a lower level of transfection of germ cells was observed than with the mRNA. It is worth noting that after one week, the reporter proteins synthesized from injected EEV were only discernible in the Sertoli cells.

Based on whole testes fluorescence and, for the first time, in vivo bioluminescence imaging of testes, we characterized the kinetics of mRNA expression. The signal measured is the fluorescence or the bioluminescence emitted by the GFP or luciferase. This signal is dependent on both the half-lives of the plasmid/mRNA and the proteins. Therefore, the kinetics of the signal persistence is a combination of the persistence of the vector and the synthesized protein. Although naked mRNAs are generally short-lived in somatic cells, several factors likely explain the prolonged detection of reporting proteins. RT-qPCR analysis showed that mRNA-GFP persisted for up to 60 hr in HEK cells and two weeks in seminiferous tubule cells (*Figure 5*), indicating enhanced stability in the testicular environment. Codon optimization and chemical modifications (capping, polyadenylation) are known to improve mRNA stability and translation efficiency (*Liu and Wang, 2022*; *Presnyak et al., 2015*; *Karikó et al., 2008*; *Pardi et al., 2018*).

The persistence of the reporter proteins is in line with the fact that proteins involved in spermatogenesis exhibit a markedly low turnover rate (*Hermann et al., 2018*). This is due to the fact that these proteins are stored within sperm organelles, such as the acrosome, manchette, centrioles, or fibrous sheath. These organelles, made during spermiogenesis, remain stable for weeks until the fertilization process occurs because there is no protein synthesis in mature sperm. For example, the Ca$_v$3.2 calcium channel is expressed during meiosis at the pachytene stage and contributes to calcium signaling during acrosome reaction (*Arnoult et al., 1999*; *Escoffier et al., 2007*; *Saunders et al., 2002*; *Saunders et al., 2007*).

When using the EEV vector, expression persisted for longer – up to 119 days – due to the intrinsic property of the EEV plasmid which allows its replication in synchronous manner with the host genome.

These results suggest that although EEV expression lasted longer, mRNAs, by targeting more efficiently male germ cells and allowing higher transfection yields of seminiferous tubules, could be a more effective and potent tool to express exogenous proteins in germ cells. By expressing a missing protein in the case of male infertility due to monogenic causes, it could be possible to restore failed spermatogenesis and thus to treat infertility.

## ARMC2 is expressed in late spermatogenesis stages

We show that ARMC2 was localized in the flagellum of spermatids obtained by enzymatic dissociation. No *Armc2* expression was detected in earlier germ cell type lineages like spermatogonia or spermatocytes. These results suggest that the ARMC2 protein is expressed late during spermatogenesis, which explains why motile sperm were found in the cauda epididymis from 3 weeks after

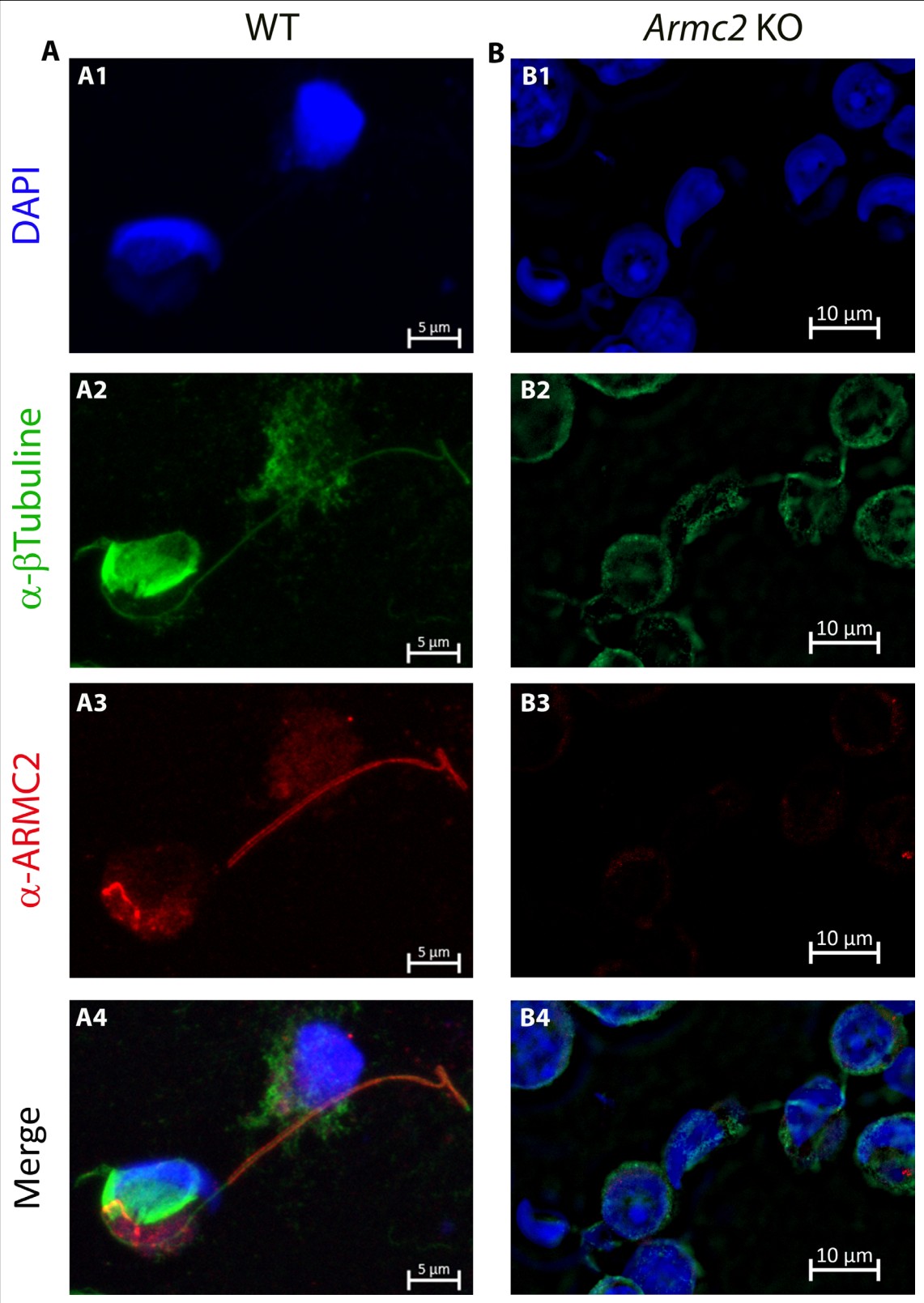

**Figure 11.** ARMC2 localization in dissociated testicular cells observed by immunofluorescence. Cells from WT and *Armc2* KO mice were counterstained with Hoechst (**A1–B1**) and stained with antibodies against tubulin (**A2–B2**, green signal) and ARMC2 (**A3–B3**, red signal). (**A4–B4**) Overlay of the different staining. In WT mice, ARMC2 is located in the flagellum of spermatids. In KO mice, no ARMC2 signal (red fluorescence) was observed in any cells.

*Figure 11 continued on next page*

*Figure 11 continued*

The online version of this article includes the following figure supplement(s) for figure 11:

**Figure supplement 1.** *Armc2* expression and localization in mice testis.

injection in our treated mice. Indeed, full spermiogenesis (from round cells to sperm) takes around 15 days (*Ibtisham, 2017*), and the journey across the epididymis lasts around 8 days, making a total of 3 weeks. Our results also confirm those recently published by *Lechtreck et al., 2022* from their study of the role of ARMC2 in the Intra-Flagellar Transport (IFT) of radial spokes in *Chlamydomonas*. They suggested that the transport of the radial spokes along the flagellum involves ARMC2, acting as an IFT adapter (*Lechtreck et al., 2022*). The presence of ARMC2 in the flagella of elongating spermatids supports this hypothesis.

## Exogenous *Armc2*-mRNA expression rescued the motility of oligo-astheno-teratozoospermic sperm

This is the first demonstration that proteins can be expressed in the testis following electroporation with optimized, capped, and poly(A)-tailed mRNA.

Our objective was to develop a new targeted therapeutic approach for infertility associated with monogenic defects. The objective of this preclinical study was to ascertain the efficacy of mRNA in expressing a missing protein, ARMC2, in a mouse model exhibiting oligo-astheno-teratozospermia due to the absence of *Armc2*, with the aim of restoring flagellar motility and fertility. Our results strongly suggest that the strategy did not alter spermatogenesis, as injection and electroporation of *Armc2*-mRNA or EEV-*Armc2* had no effect on testicular morphology or weight. More importantly, the technique was effective, with motile sperm cells found in cauda epididymis 3 and 5 weeks after *Armc2*-mRNA injection into testes from *Armc2* KO males. Nevertheless, it should be noted that not all injected mice were efficiently treated. For example, only 87.5% of the treated mice (14 of the 16) exhibited motile sperm after 5 weeks. The absence of motile sperm at 5 weeks may be attributed to the specific types of spermatogenic cells that were transfected during the electroporation phase. It is possible that the transfected cells may differ between individuals, potentially influenced by the injection and the position of the electrodes during electroporation. If the mRNA transfection occurs in a spermatogonia, it may take more than six weeks (including the time required to cross the epididymis) before motile epididymal sperm cells emerge. This potential timeline could explain the absence of motile sperm in some mice at 5 weeks.

Due to the quantity of motile sperm obtained, it was not possible to produce offspring through natural mating. Nevertheless, the rescued *Armc2*$^{-/-}$ sperm displayed normal morphology and motility and were competent for IVF and ICSI. The rescued sperm successfully fertilized oocytes and produced embryos that developed to the 2-cell stage following IVF and to the blastocyst stage after ICSI. These findings support a partial functional recovery of the sperm population. Taken together, these results demonstrate that mRNA electroporation can partially restore sperm function by rescuing motility and enabling early embryonic development. This proof-of-concept supports the feasibility of mRNA-based protein supplementation in spermatogenic cells, paving the way for future therapeutic development.

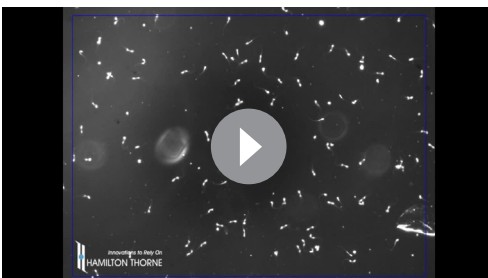

**Video 3.** CASA recording of WT epididymal sperm cells.

https://elifesciences.org/articles/94514/figures#video3

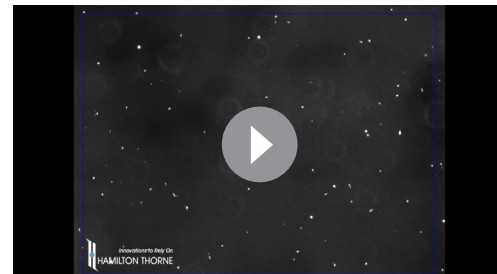

**Video 4.** CASA recording of *Armc2* KO epididymal sperm cells.

https://elifesciences.org/articles/94514/figures#video4

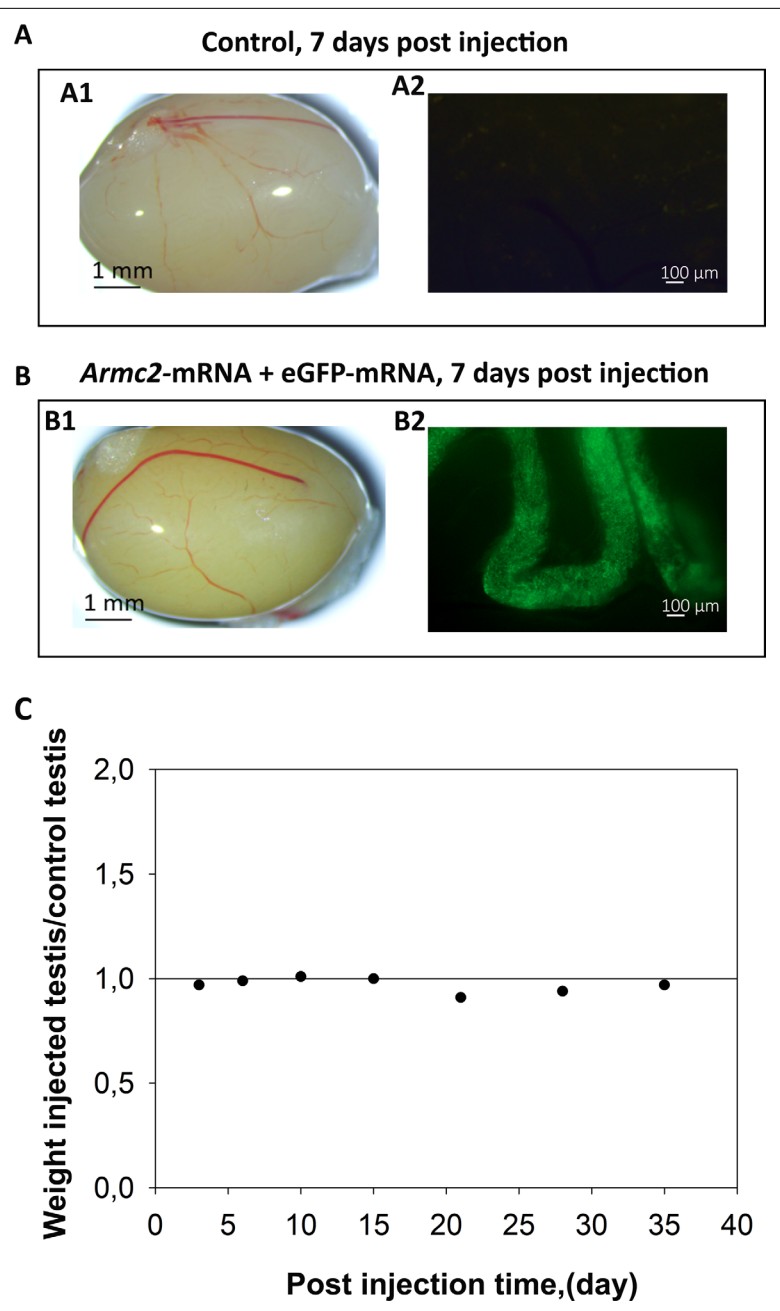

**Figure 12.** In vivo co-injection of *Armc2*-mRNA and *eGFP*-mRNA followed by electroporation does not affect testes morphology and weight. Adult WT mouse testes were injected with a solution containing *Armc2*-mRNA and *eGFP*-mRNA. After injection, the testes were electroporated and mice were euthanized two weeks later. (**A**) Whole testis under white and blue lights on a fluorescence microscope. (**A1**) Control testes not injected/electroporated. (**A2**) Testes injected with *Armc2*-mRNA and *eGFP*-mRNA. *eGFP*-mRNA was co-injected to follow the transfection efficiency. (**B**) Ratio of injected/electroporated testis weights to control testis weights at several time points post-injection (3-, 6-, 10-, 15-, 21-, 28-, and 35-day post-surgery). $n = 1$ mouse per time.

The online version of this article includes the following source data and figure supplement(s) for figure 12:

**Figure supplement 1.** Validation of *Armc2*-mRNA in HEK cells.

**Figure supplement 1—source data 1.** PDF file containing original western blots for *Figure 12—figure supplement 1D*, indicating the relevant bands.

**Figure supplement 1—source data 2.** Original files for western blot analysis displayed in *Figure 12—figure supplement 1D*.

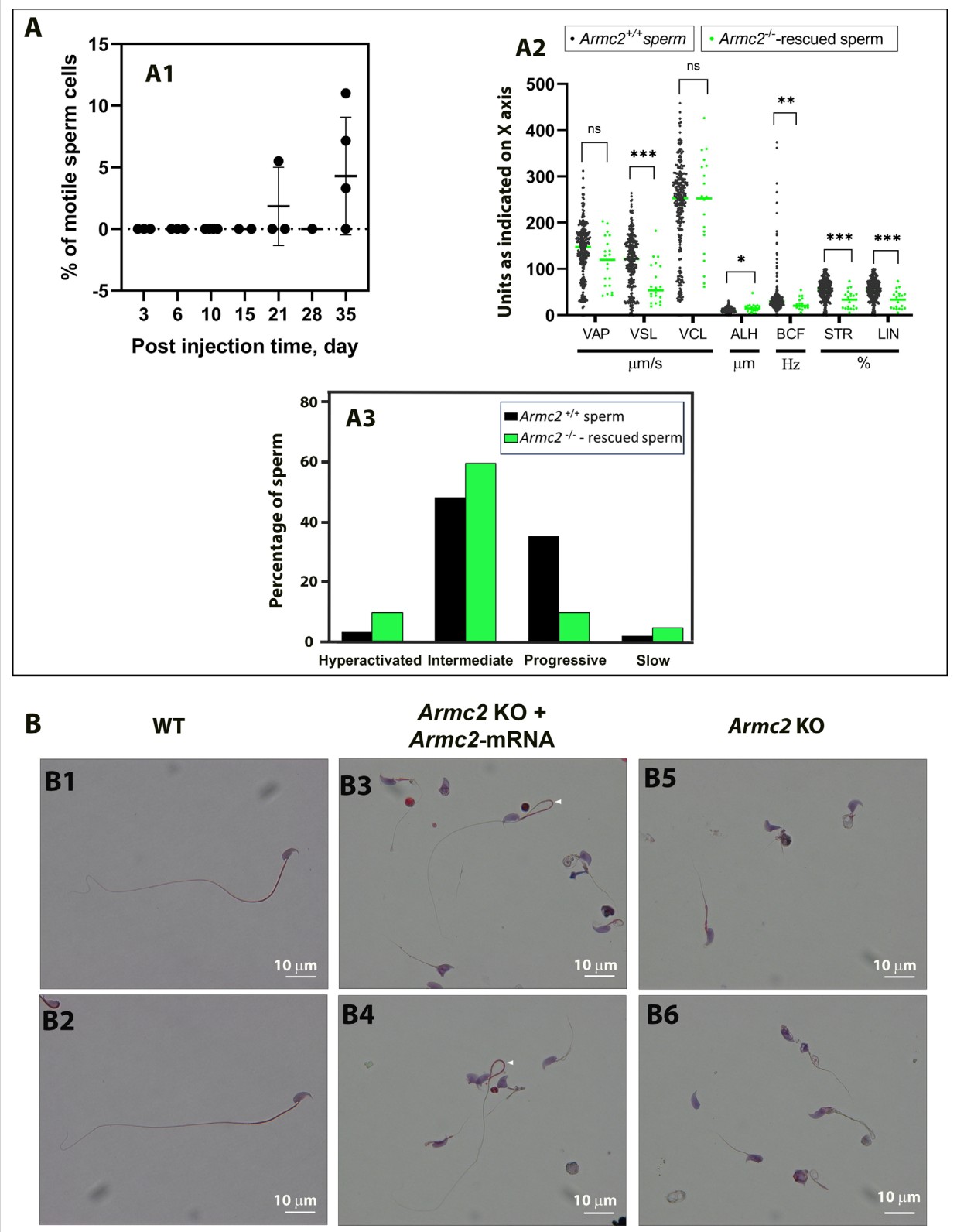

**Figure 13.** Sperm motility is restored in *Armc2* KO mice at 21 and 35 days after injection and electroporation of *Armc2*-mRNA. (**A**) Adult *Armc2* KO mouse testes were injected with a solution containing *Armc2*-mRNA. After injection, the testes were electroporated. At different times (3-, 6-, 10-, 15-, 21-, 28-, and 35-day *post*-injection), sperm were extracted from the cauda epididymis of the injected testis, and the sample was then examined with a CASA system to identify the percentage of motile spermatozoa (**A1**). *n* = 2 for day 15, *n* = 3 for days 3, 6, and 21, *n* = 4 for day 10, and *n* = 5 for days

*Figure 13 continued on next page*

*Figure 13 continued*

28 and 35. (**A2**) Sperm motility parameters of *Armc2⁻/⁻*-rescued sperm in comparison to *Armc2⁺/⁺* sperm. The motility parameters measured were: averaged path velocity (VAP); straight line velocity (VSL); curvilinear velocity (VCL); amplitude of lateral head displacement (ALH); beat cross frequency (BCF); straightness (STR); linearity (LIN). Black dots: sperm cells from *Armc2* null mice, green dots: sperm cells from *Armc2* null mice 35 days after injection with *Armc2*-mRNA. Results are expressed as mean ± SD. (**A3**) Sperm motility population of *Armc2⁻/⁻*-rescued sperm in comparison to *Armc2⁻/⁻* sperm. Black column: sperm cells from *Armc2* null mice, green column: sperm cells from *Armc2* null mice 35 days after injection with *ARmc2*-mRNA. Statistical significance was verified using a Mann–Whitney sum test. Data are displayed as mean ± SEM. p values of *≤0.05, **≤0.01, or ***≤0.001 were considered to represent statistically significant differences. (**B**) Morphology of sperm cells in *Armc2* KO mice injected or not with *Armc2*-mRNA. (**B1, B2**) Microscopic observation of epididymal sperm cells from a mature WT mouse. (**B3, B4**) Epididymal sperm cells from a mature *Armc2* KO mouse 35 days after injection/electroporation with *Armc2*-mRNA. (**B5, B6**) Epididymal sperm cells from a control *Armc2* KO male. Normal sperm cells were observed in the injected condition with *Armc2*-mRNA (white arrows). Scale bars: 10 μm.

The online version of this article includes the following figure supplement(s) for figure 13:

**Figure supplement 1.** Morphology of sperm cells from WT, *Armc2* KO, and *Armc2*-rescued males.

---

It is worth noting that the significant modifications of the CASA parameters observed for rescued sperm did not impact their fertilizing potential. Naked mRNA injection/electroporation is therefore a promising method to treat infertility. In contrast, no motile spermatozoa were found after injection/electroporation of EEV-*Armc2*, confirming our previous results suggesting that this nucleic tool does not efficiently enter or transfect germ cells.

Despite this success, the transfection rate deserves to be improved to obtain a larger number of sperm cells to produce offspring through natural mating. To increase the testicular transfection rate, encapsulation of mRNA into lipid nanoparticles could be used, as used for Covid vaccination (*Schoenmaker et al., 2021*). During the writing of this manuscript, the Dong team (*Du et al., 2023*) used a self-amplifying RNA (saRNA) encapsulated in cholesterol-amino-phosphate derived lipid nanoparticle to restore spermatogenesis in infertile mice. They successfully restore the expression of the DNA Meiotic Recombinase 1 (DMC1) (*Takemoto et al., 2020*; *Habu et al., 1996*; *Shinohara et al., 1997*) in *Dmc1* KO infertile mice by injecting a self-amplifying RNA-*Dmc1* in the testes. saRNA are genetically engineered replicons derived from self-replicating single-stranded RNA viruses (*Tews et al., 2017*). The saRNA contains the alphavirus replicase genes and encodes an RNA-dependent RNA polymerase (RdRP) complex which amplifies synthetic transcripts in situ and the target RNA sequence. The target RNA is expressed at high levels as a separate entity. As a result of their self-replicative activity, saRNAs can be delivered at lower concentrations than conventional mRNA to achieve comparable expression levels (*Bloom et al., 2021*). Moreover, saRNA constructs need to be condensed by a cationic carrier into a nanoparticle measuring ~100 nm to enable their uptake into target cells and protect the saRNA from degradation (*Kim et al., 2021*). Finally, saRNA will amplify the RNA without cellular regulation. For all these reasons, if such a strategy is to be pursued, a potential toxicity effect due to saRNA overexpression must be investigated in the testes and progeny. Another difficulty with saRNA relates to the size of the molecular construct. saRNA sequences are large and complex. The length of the sequence RdRP is around 7 kb, which often makes the full length of saRNA more than 9 kb once the sequence for the protein of interest has been integrated (*Kim et al., 2021*). *Du et al., 2023* successfully used their saRNA construct to rescue spermatogenesis failure induced by the absence of the

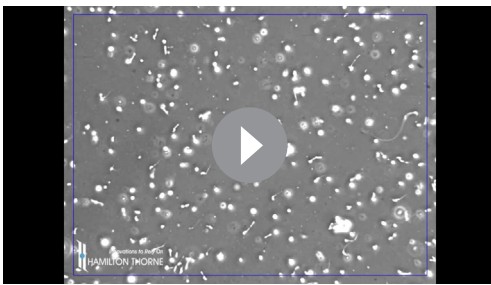

**Video 5.** CASA recordings of epididymal sperm cells from *Armc2* KO mice on days 21 and 35, respectively, after injection/electroporation with *Armc2*-mRNA.
https://elifesciences.org/articles/94514/figures#video5

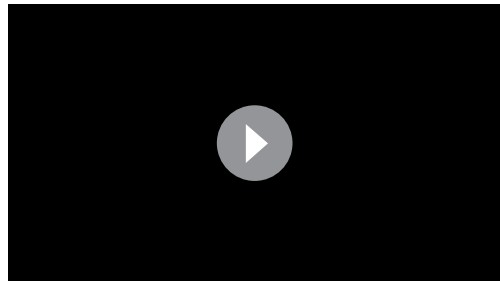

**Video 6.** CASA recordings of epididymal sperm cells from *Armc2* KO mice on days 21 and 35, respectively, after injection/electroporation with *Armc2*-mRNA.
https://elifesciences.org/articles/94514/figures#video6

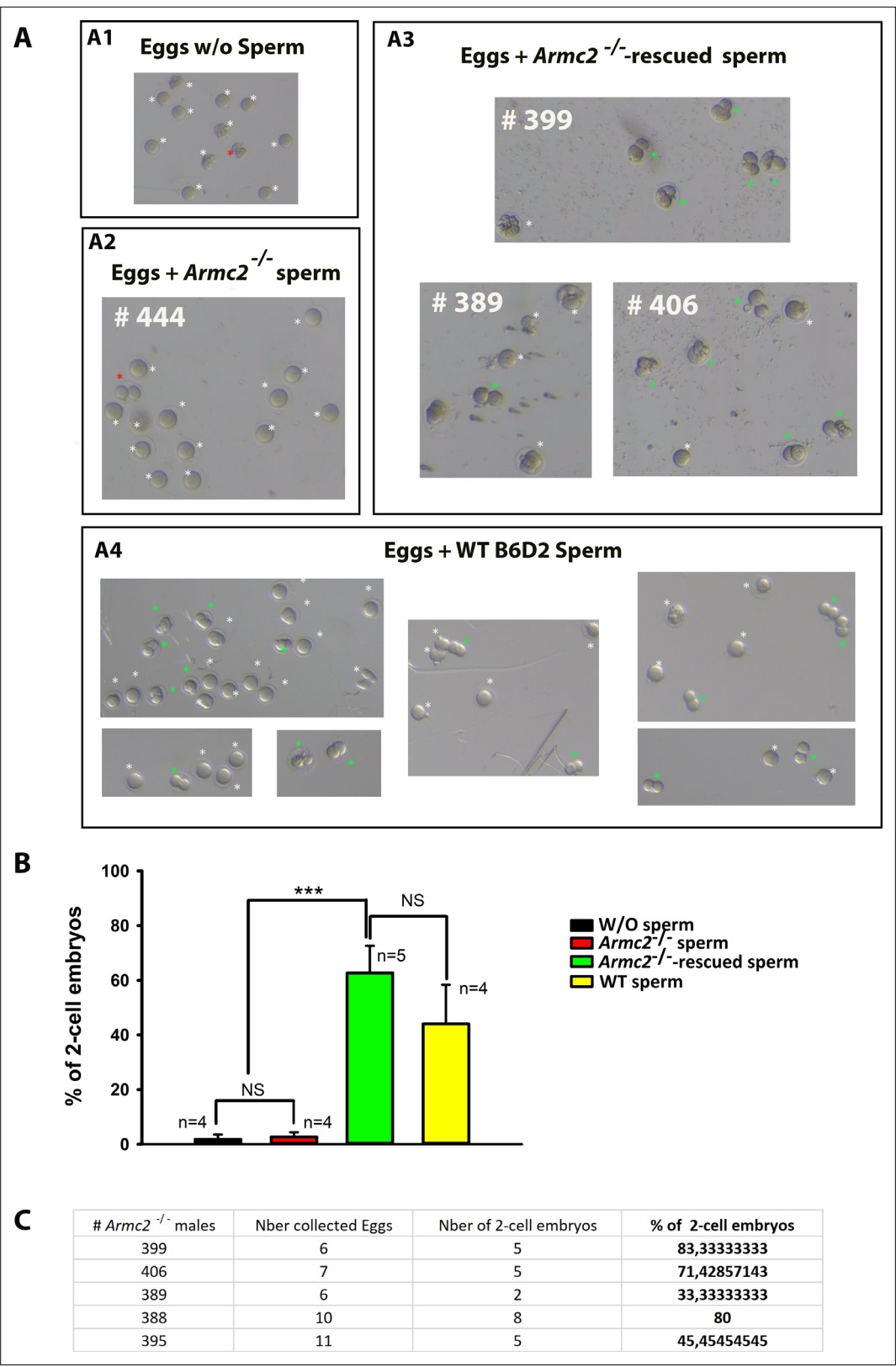

**Figure 14.** *Armc2⁻/⁻*-rescued sperm can fertilize eggs and produce embryos by in vitro fertilization (IVF).
(**A**) Illustrations showing eggs/embryos obtained 24 hr after egg collection in the following conditions. (**A1**) W/O sperm, (**A2**) IVF with *Armc2⁻/⁻* sperm from *Armc2⁻/⁻* males (#444), (**A3**) IVF with *Armc2⁻/⁻* rescued sperm from *Armc2⁻/⁻* males treated with mRNA-*Armc2* (males #399, #389, and #406), and (**A4**) IVF with WT B6D2 sperm. Green

*Figure 14 continued on next page*

*Figure 14 continued*

and red asterisks show 2-cell embryos, red asterisks showing 2-cell embryos obtained by parthenogenesis. White asterisks show unfertilized oocytes or degenerated. (**B**) Histograms showing the mean percentage ± SD of alive eggs reaching the 2-cell embryo stage at 24 hr after IVF without sperm (*n* = 4), with *Armc2*⁻/⁻ sperm (*n* = 4), with *Armc2*⁻/⁻ rescued sperm (*n* = 5), and with WT B6D2 sperm (*n* = 5). Statistical significance was verified using a one-way ANOVA test. (**C**) The table presents the data pertaining to the number of 2-cell embryos obtained by IVF with *Armc2*⁻/⁻ rescued sperm from five different *Armc2*⁻/⁻ males treated with mRNA-*Armc2* (males #399, #406, #389, #388, and #395). The data is presented in terms of the percentage of 2-cell embryos obtained in relation to the total number of eggs collected.

The online version of this article includes the following figure supplement(s) for figure 14:

**Figure supplement 1.** *Armc2*⁻/⁻-rescued sperm can fertilize eggs and produce embryos by ICSI.

---

small protein Dmc1 (37 kDa), but it may be more challenging with larger proteins such as the structural proteins involved in OAT, including the 98 kDa ARMC2 (*Coutton et al., 2019*).

Our next step will be to assess whether encapsulating *Armc2*-mRNA in LNP-CAP could allow a larger number of germ cells to be transfected.

## Naked mRNA, a new therapeutic strategy to treat severe infertility

Non-obstructive azoospermia (NOA) and severe oligozoospermia (SO) are the most severe disorders of spermatogenesis and are the most likely to be of genetic origin. NOA is defined by the complete absence of spermatozoa in the ejaculate. Approximately 10–15% of infertile men have azoospermia, and a further 15% have SO (*Cocuzza et al., 2013*). For patients with NOA, few clinical solutions are currently available. Generally, testicular sperm extraction is attempted to collect some spermatozoa from the seminiferous tubules, which can then be used for ICSI (*Ma et al., 2019*). When no sperm are retrieved, intra-conjugal conception is impossible. The results of this study strongly suggest that transient mRNA expression of a missing protein in NOA testes by electroporation could be sufficient to produce normal sperm for IVF and obtain embryos.

In conclusion, this work provides the first demonstration that in vivo testicular injection and electroporation of capped and poly(A)-tailed mRNA can efficiently transfect male germ cells and produce proteins with sufficient stability to support spermatogenic progression. Our comprehensive analysis revealed that mRNA electroporation is more efficient than episomal vector delivery, despite shorter persistence of expression. Importantly, *Armc2*-mRNA electroporation was able to partially rescue sperm motility and fertilizing ability in *Armc2*⁻/⁻ mice, representing a crucial first step toward developing mRNA-based strategies to correct monogenic causes of male infertility.

## Materials and methods

**Key resources table**

| Reagent type (species) or resource | Designation | Source or reference | Identifiers | Additional information |
| --- | --- | --- | --- | --- |
| Strain, strain background (*Mus musculus*) | B6D2 F1 hybrid (♀ C57BL/6JRj × ♂ DBA/2) | Janvier laboratories | | Adult male mice, 8–25 weeks old |
| Strain, strain background (*Mus musculus*) | CD1 female | Janvier laboratories | | 6–7 weeks old, used for IVF/ICSI |
| Genetic reagent (*Mus musculus*) | *Armc2* KO | CRISPR–Cas9 | | Adult male mice, 8–15 weeks old; ***Coutton et al., 2019*** |
| Cell line (*Homo sapiens*) | HEK293T | ATCC | CRL-3216; RRID:CVCL_0063 | Transfection experiments with EEV-Armc2 and mRNA |
| Biological sample (*Mus musculus*) | Testicular cells and sperm | | | Freshly isolated from *Mus musculus* |

*Continued on next page*

*Continued*

| Reagent type (species) or resource | Designation | Source or reference | Identifiers | Additional information |
|---|---|---|---|---|
| Biological sample (*Mus musculus*) | Egg | | | Freshly isolated from *Mus musculus* |
| Antibody | anti-HA (Rabbit polyclonal) | Sigma-Aldrich | 11867423001;RRID: AB_390916 | 1:5000 for Western blot |
| Antibody | anti-tubulin (Guinea pig polyclonal) | Swiss Institute of Bioinformatics | AA345; RRID:AB_2139275 | 1:100 for immunofluorescence |
| Antibody | anti-ARMC2 (Rabbit polyclonal) | Sigma-Aldrich | HPA053696; RRID:AB_2679152 | 1:50 for immunofluorescence |
| Recombinant DNA reagent | mCherry plasmid | Dr. Conti, UCSF | | CMV/T7 promoter; amplified in *E. coli*, (*Homo sapiens*) |
| Sequence-based reagent | *Armc2*_WT Forward; *Armc2*_WT Reverse | Genscript | PCR primers | For genotyping *Armc2* WT allele (*Mus musculus*) |
| Sequence-based reagent | *Armc2*_KO Forward; *Armc2*_KO Reverse | Genscript | PCR primers | For genotyping *Armc2* KO allele (*Mus musculus*) |
| Sequence-based reagent | *Gfp* Forward; *Gfp* Reverse | Genscript | PCR primers | For GFP expression analysis in seminiferous tubules (*Mus musculus*) |
| Sequence-based reagent | *Gadph* Forward; *Gadph* Reverse | Genscript | PCR primers | For qPCR normalization in HEK cells (*Homo sapiens*) |
| Commercial assay or kit | NucleoBond Xtra Midi kit | Macherey-Nagel | 740410-50 | For plasmid purification |
| Chemical compound, drug | Ketamine | Centravet | | Anesthesia for mice |
| Chemical compound, drug | Xylazine | Centravet | | Anesthesia for mice |
| Chemical compound, drug | Fast Green | Sigma-Aldrich | F7258 | 0.05% in injections |
| Software, algorithm | Imaris | Oxford Instruments | | 3D reconstruction of cleared testes |
| Software, algorithm | FIJI | Open-source | | Image analysis |
| Other | *Armc2*-mRNA | Trilink | | CleanCap AG, poly(A) 120, 3 HA-FLAG; injected 300 ng/µl (*Mus musculus*) |
| Other | *EGFP*-mRNA | Trilink | | Used as reporter; co-injected with *Armc2*-mRNA (*Mus musculus*) |
| Other | *mCherry*-mRNA | Trilink | | Transcribed in vitro; capped and polyadenylated; 300 ng/µl (*Homo sapiens*) |

## Animals

All procedures involving animals were performed in line with the French guidelines for the use of live animals in scientific investigations. The study protocol was approved by the local ethics committee (ComEth Grenoble # 318) and received governmental authorization (ministerial agreement # 38109-2022072716142778).

The animals used were (1) B6D2 F1 hybrid (Å C57BL/6JRj crossed with Å DBA/2, Janvier laboratories; France) adult male mice aged between 8 and 25 weeks, (2) the *Armc2* KO mouse strain obtained by CRISPR–Cas9 (*Coutton et al., 2019*), and (3) CD1 female 6 weeks old (Janvier laboratories, Le Genest-Saint-Isle, France). Experiments were carried out on wild type (WT) or *Armc2* KO adult male mice aged between 8 and 15 weeks.

## Chemicals and reagents

Fast Green (FG) (F7258 – 25 g), phosphate-buffered saline (PBS, D853 7–500 ml), hematoxylin (GH5316 – 500 ml), eosin (HT110216 – 500 ml), tert-butanol (471712 – 100 ml), Histodenz (D2158 – 100 g), sorbitol (S1876 – 100 g), urea (U5128 – 500 g), potassium phosphate, monobasic (P0662), magnesium sulfate (anhydrous) (M9397), sodium bicarbonate calcium chloride · 2H$_2$O (22317), EDTA (E9884), sodium lactate (60% syrup – d = 1.32 g $^{l-1}$) (L7900), glucose (G8270), penicillin (P4333), streptomycin (P4333), HEPES (H0887), PVA 30,000–70,000 (P8136), albumin, bovine fraction V (A3803),

RPMI-1640 media (R2405), M2 medium (M7167), hyaluronidase (H3884), mineral oil (M8410), Poly-VinylPyrolidone (PVP360-100G), M16 medium (MR-016), trypsin (T6567), and Collagenase (C8176) were purchased from Sigma-Aldrich (Saint-Quentin-Fallavier, France). Sodium chloride (1112 A) was purchased from Euromedex (Souffelweyersheim, France). Potassium chloride (26764), L-glutamine (35050038), Hoechst 33342 (R37165), sodium pyruvate (11360039), KSR serum (10828010), NEAA (11140050), and EAA (11130036) were purchased from Life Technologies (Waltham, MA USA). Schorr staining solution was obtained from Merck (Darmstadt, Germany). Mfel HF (R35895) and RNase-free DNase I (M03035) were obtained from New England Biolabs (Ipswich, MA, USA). Paraformaldehyde (PFA, 15710) was obtained from Electron Microscopy Science (Hatfield, PA, USA). Ketamine and xylazine were obtained from Centravet (Dinan, France). Fluorescent i-particles (NIRFiP-180) were obtained from Adjuvatis (Lyon, France). Script Guard RNAse CleanCap AG (040N-7113-1), CleanCAp *EGFP*-mRNA (040L-7601-100), T7-FlashScribe kit (C-ASF3507), poly(A) tail (C-PAP5104H) and CleanCap *Luciferase*-mRNA (L-7602–1000) were obtained from Tebubio (Le Perray en Yvelines, France). Promega Blue Orange G1904, Promega, (Charbonnières France).

## Plasmids

All plasmids, EEV *CAGs-GFP-T2A-Luciferase* (EEV604A-2; System Bioscience, Palo Alto, CA, USA), mCherry plasmid (given by Dr. Conti MD at UCSF, San Francisco, CA, USA), and EEV-*Armc2-GFP* plasmid (CUSTOM-S017188-R2-3, Trilink, San Diego, CA, USA) were amplified by bacterial transformation (*E. coli*, EC0112; Thermo Fisher, Courtaboeuf, France). After expansion, plasmids were purified with a NucleoBond Xtra Midi kit (740410-50; Macherey-Nagel, Düren, Germany) using manufacturer's protocol. All plasmid DNA pellets were solubilized in (DNAse- and RNAse-free) milliQ water (Sigma-Aldrich, Saint-Quentin-Fallavier, France), before being used.

The EEV *CAGs-GFP-T2A-Luciferase* episome contains the cDNA sequences of Green Fluorescent Protein (GFP) and luciferase, under the control of a CAGs promoter (*Figure 1—figure supplement 1*). After purification, the EEV *CAGs-GFP-T2A-Luciferase* plasmid concentration was adjusted to 9 µg µl$^{-1}$. Prior to injection, 3.3 µl of this plasmid solution was mixed with 1 µl 0.5% Fast Green and 5.7 µl sterile PBS to obtain a final EEV concentration of 3 µg µl$^{-1}$. The EEV-*Armc2-GFP* plasmid contains the mouse cDNA sequences of *Armc2* (*ENSMUST00000095729.11*) and the GFP genes under the control of a strong CAGs promoter (*Figure 1—figure supplement 1*). After amplification and purification, the final plasmid concentration was adjusted to 9 µg µl$^{-1}$ in water. Prior to injection, 3.3 µl of this plasmid solution was mixed with 1 µl of 0.5% Fast Green and 5.7 µl of sterile PBS to obtain a final EEV concentration of 3 µg µl$^{-1}$. The *mCherry* plasmid contains the cDNA sequence of *mCherry* under the control of CMV and T7 promoters (*Figure 1—figure supplement 1*). After amplification and purification, the final plasmid concentration was adjusted to 9 µg µl$^{-1}$.

## *Armc2*-mRNA

*Armc2*-mRNA used for in vivo testes microinjection and electroporation was obtained from Trilink (San Diego, CA, USA). The commercial *Armc2*-mRNA has an AG CleanCap, a poly (A) tail of 120 adenosines and 3 HA-FLAG. The main challenge with mRNA-based therapy is mRNA stability. To improve mRNA stability in vivo and avoid its degradation by ribonucleases, optimization techniques were implemented. Thus, the used mRNA has codon optimization, a poly (A) tail, and a CleanCap (*Wu et al., 2019*; *Gallie, 1991*; *Henderson, 2021*; *Stepinski, 2001*; *Sachs et al., 1997*). To verify the efficiency of cell transfection, an *EGFP*-mRNA was injected together with the *Armc2*-mRNA. During the injection, the concentration of *EGFP*-mRNA and *Armc2*-mRNA was 300 ng µl$^{-1}$ each.

## *mCherry*-mRNA transcription in vitro

The circular *mCherry* plasmid was linearized using the restriction enzyme Mfel HF at 37°C for 1 hr. The pm-*mCherry* was then extracted and purified with the DNA Clean and Concentrator-25 kit (D4033; Zymo Research, Irvine, CA, USA). The pm-*mCherry* was subsequently transcribed in vitro using the T7-FlashScribe kit (C-ASF3507; Tebubio, Le Perray en Yvelines, France). The mRNA was capped with a clean cap (CleanCap AG; 040N-7113-1, Tebubio, Le Perray en Yvelines, France), and a poly(A) tail (C-PAP5104H; Tebubio, Le Perray en Yvelines, France) was added before purification using the NucleoSpin RNA Clean Up kit (740948-50; Macherey-Nagel, Düren, Germany). Prior to injection, *mCherry*-mRNA was mixed with Fast Green to obtain a final concentration of 300 ng µl$^{-1}$ (0.05% FG, PBS).

## Agarose gel electrophoresis of the Episomal Vector EEV and mRNA-mCherry

Before loading on a pre-stained (Gel Green) 1.5% agarose gel, the EEV-plasmid and mRNA were mixed with a loading dye (Promega Blue Orange G1904, Promega, Charbonnières France). An aliquot of each sample (500 ng) was loaded into each well and electrophoresis was performed for 30 min at 100 V at room temperature (RT). A DNA size marker (Gene ruler SM1331, Thermo Scientific, Courtaboeuf, France) was used to assess molecular weight. Gel images were acquired using a ChemiDoc XRS+ (Bio-Rad, Marnes-la-Coquette, France).

## In vivo microinjection and electroporation of testes

Electroporation was conducted as previously described (*Michaelis et al., 2014*). Briefly, male mice B6D2, *Armc2*[+/+], or *Armc2*[-/-], depending on the experimental conditions, were anesthetized with ketamine/xylazine solution (100 and 10 mg µl$^{-1}$, respectively). The testes were pulled out of the abdominal cavity or scrotum. Under a binocular microscope and using microcapillaries pipettes (FemtoTip II, Eppendorf, Montesson, France), 10 µl DNA (3 µg µl$^{-1}$ – 0.05% FG) or mRNA (300 ng µl$^{-1}$ – 0.05% FG) was injected into the seminiferous tubules through the *rete testis* applying a constant pressure (microinjector, Femto Jet 4i, Eppendorf, Montesson, France). Two series of 8 square electric pulses (25 V for 50ms) were applied to the testis using tweezer-type electrodes linked to an electroporator (Gemini, BTX, Holliston, MA, USA). The testes were then replaced in the abdominal cavity, and the abdominal wall and skin were sutured. For each animal, the left testis was injected and electroporated with the different nucleic acids (mRNA, EEV), whereas the right testis was injected with a control solution (PBS, 0.5% FG) as a control. Both testes were electroporated.

## Flash freezing and fluorescence analysis of testes

Depending on the experimental condition, 1-, 3-, or 7-day post-injection, the testes were collected and washed for 5 min in PBS. Then, they were embedded in Peel-A-Way Cryomolds filled with OCT Mounting media (VWR, Rosny-sous-Bois, France). The samples were flash frozen in a 100% isopentane solution (524391; Carlo ERBA, Val-de-Reuil, France), pre-cooled with liquid nitrogen. Once frozen, they were cut into 20 µm sections using a cryostat. The cryostat sections were then fixed with 4% PFA-PBS for 10 min at 4°C and counterstained with 1.8 µM DAPI (nuclear stain) before observation using an Axioscan Z1 slide scanner or a confocal microscope LSM710 (NLO – LIVE7 – Confocor 3). The fluorescence of the different reporter proteins was detected using appropriate filters for DAPI, GFP, Texas Red (for mCherry), and Cy7 (for the Fluorescent i-particles NIRFiP-180).

**Table 1.** Primer sequences used for mouse genotyping and *Gfp* expression analysis in HEK cells and mouse seminiferous tubules.

| | | Primer sequences (5'–3') | Final concentration (µM) |
|---|---|---|---|
| Genotyping | | | |
| *Armc2*_WT | Forward | GGCCCGAGCACGCTTCTA | 0.4 |
| | Reverse | TTCATGTAAGAACTATCCAGGACCA | 0.4 |
| *Armc2*_KO | Forward | TGGGACGCAGCCCTGTAA | 0.4 |
| | Reverse | AACCCAAAGCTCCAGCATCTC | 0.4 |
| Expression | | | |
| *Gfp* | Forward | ACGACTTCTTCAAGTCCGCC | 0.08 |
| | Reverse | TCTTGTAGTTGCCGTCGTCC | 0.08 |
| *Gadph* Human | Forward | TCTCTGCTCCTCCTGTTCGA | 0.33 |
| | Reverse | TTCCCGTTCTCAGCCTTGAC | 0.33 |
| *Gadph* mice | Forward | | |
| | Reverse | | |

## Tissue collection and histological analysis

For histological analysis, treated and control B6D2 testes were fixed by immersion in 4% PFA for 14 hr. They were then embedded in paraffin before cutting into 5 µm sections using a microtome (Leica biosystems, Wetzlar, Germany). After deparaffination, the sections were stained with hematoxylin and eosin. Stained sections were digitized at ×20 magnification using an Axioscan Z1-slide scanner (Zeiss, Jena, Germany). Spermatogenesis was assessed by measuring the area of seminiferous tubules and the cross sections of round tubules ($\mu m^2$) ($n>35$ tubules per testis section; $n = 5$ testis sections per condition). Statistical significance of differences was determined using a Student's $t$-test.

## Ex vivo fluorescence analysis

To analyze the expression of the reporter proteins GFP and mCherry in seminiferous tubules, whole testes were examined under an inverted microscope (CKX53, Olympus, Shinjuku, Tokyo, Japan). Exogenous fluorescence was detected using filters for GFP and Texas Red.

## *Armc2* genotyping

DNA was obtained by the digestion of an ear punch using the Direct PCR lysis buffer (Euromedex, Souffelweyersheim, FR) and 1 mg/ml proteinase K (Thermo Fisher Scientific, Waltham, Massachusetts, USA) overnight at 55°C, followed by enzyme inactivation 1 hr at 85°C. PCR was performed as follows: 2 µl of undiluted DNA was mixed with 0.4 µM primers against *Armc2* gene (Genscript, Piscataway, NJ, USA, sequences are listed in *Table 1*), Coral Load PCR Buffer 1X (QIAGEN, Hilden, DE), 0.8 mM dNTP mix (Thermo Fisher Scientific) and 0.625 units of Taq DNA Polymerase (QIAGEN) in a final volume of 25 µl. The PCR program used was 94°C 15 min (94°C 30 s, 59°C 30 s, 72°C 1 min) × 32 followed by a final elongation (72°C 10 min). PCR products were then visualized by electrophoresis (Bio-Rad, Hercules, California, USA) on a 1% agarose gel with Gelgreen 1,8X (Interchim, Montluçon, FR).

## Purification of seminiferous tubule cells

Depending on the experimental condition, testes were collected at 1, 3-, 7-, 14-, or 21-day post-injection of mRNA-GFP (300 ng/µl) and transferred into a sterile 35 mm Petri dish containing 2 ml of 1× PBS (pH 7.4). The tunica albuginea was carefully removed under sterile conditions. Testicular tissue was then mechanically dissociated by gently aspirating and expelling the tissue 10–12 times using an 18-gauge needle to isolate individual seminiferous tubules. To enrich for GFP-positive tubules, tubules lacking GFP fluorescence were discarded under a fluorescence microscope. This procedure resulted in a sample containing approximately 25% GFP-positive tubules.

The isolated tubules were washed three times with PBS to remove residual debris. Enzymatic dissociation was performed in two steps: first, the tubules were incubated in collagenase (1 mg/ml in RPMI 1640 medium (Sigma-Aldrich, Saint-Louis, Missouri, USA)) for 30 min at 37°C with gentle agitation. This was followed by a second digestion step using trypsin (2 mg/ml in RPMI 1640 medium (Sigma-Aldrich, Saint-Louis, Missouri, USA)) for 15 min at 37°C with gentle agitation to ensure complete dissociation of seminiferous epithelial cells.

To stop the enzymatic digestion, cells were immediately resuspended in RPMI 1640 medium (Sigma-Aldrich, Saint-Louis, Missouri, USA) supplemented with 5% KnockOut Serum Replacement (KSR; Thermo Fisher Scientific). The resulting single-cell suspension was then used for downstream applications.

## Live mRNA-GFP fluorescence imaging of dissociated seminiferous tubule cells

The isolated seminiferous tubule cell suspensions obtained from testes 24-hr post-injection with mRNA-GFP (300 ng/µl) were cultured in RPMI 1640 medium (Sigma-Aldrich, Saint-Louis, Missouri, USA) in chambered coverglass systems (Nunc Lab-Tek II, Cat. No. 155360PK, Thermo Fisher Scientific, Courtaboeuf, France) at 37 °C in a humidified atmosphere containing 5% $CO_2$.

After 12 hr of culture, the cells were counterstained with 1.8 µM Hoechst 33342 (Thermo Fisher Scientific), a membrane-permeable nuclear dye functionally similar to DAPI. Live-cell fluorescence imaging was performed using an Axio Observer 7 inverted fluorescence microscope (Zeiss, Jena, Germany) equipped with a 63× oil immersion objective. GFP and Hoechst fluorescence signals were

detected using the appropriate filter sets. Image acquisition and analysis were conducted using ZEN lite software (Zeiss, Jena, Germany).

## Cell culture

HEK cells were cultured in 100 mm tissue culture dishes (Falcon, Corning, New York, USA) in DMEM/F12 medium (Sigma-Aldrich, Saint Louis, Missouri, USA) supplemented with 10% heat-inactivated fetal bovine serum (Life Technologies), 1% penicillin/streptomycin (Sigma-Aldrich), and 0.01 mg/ml hygromycin B (Thermo Fisher Scientific). Cells were maintained at 37°C in a humidified atmosphere containing 5% $CO_2$ and were passaged twice weekly at a 1:10 dilution.

Transfection was performed using 2 µg/ml GFP mRNA (TriLink BioTechnologies, San Diego, USA) and the jetMESSENGER mRNA transfection reagent (Sartorius, Göttingen, Germany), following the manufacturer's instructions. mRNA was extracted from the cells at 12-, 24-, 48-, and 60-hr post-transfection.

## mRNA extraction and qPCR analysis

Total RNA from isolated seminiferous tubule cell suspensions and HEK cells was extracted using the NucleoSpin RNA Plus kit (Macherey-Nagel, Düren, Germany) according to the manufacturer's protocol. RNA concentration was measured using the Qubit RNA Assay Kit (Life Technologies, Carlsbad, California, USA), and samples were stored at –80°C until further use.

Up to 800 ng of total RNA per sample was reverse-transcribed into cDNA using the iScript cDNA Synthesis Kit (Bio-Rad, Hercules, California, USA), following the manufacturer's protocol. Gene expression was analyzed by quantitative PCR (qPCR) using 1 µl of undiluted cDNA in a final reaction volume of 10 µl, with gene-specific primers (see *Table 1*), and the SsoAdvanced Universal SYBR Green Supermix (Bio-Rad, Hercules, California, USA).

The qPCR program for GAPDH expression in HEK cells included an initial denaturation at 95°C for 3 min, followed by 39 cycles of 95°C for 10 s, 60°C for 30 s, and 72°C for 30 s. Melt curve analysis was performed by increasing the temperature from 58°C to 95°C in 0.5°C increments, holding each step for 2–5 s.

For GFP expression analysis, the same program was used, except that the annealing temperature was set to 62°C instead of 60°C.

Relative gene expression was calculated using the $2^{-\Delta CT}$ method.

## Harris–Shorr sperm analysis

Sperm were collected from the caudae epididymides of mice (Control, injected with EEV-*GFP*, *GFP*-mRNA, or *Armc2*-mRNA). They were allowed to swim for 10 min at 37°C in 1 mL M2 media. Sperm were centrifuged at 500 × *g*, washed once with PBS, and fixed 4% PFA in PBS for 1 min at RT. After washing with 1 ml acetate ammonia (100 mM), the sperm suspensions were spotted onto 0.1% poly-L-lysine precoated slides (Thermo Scientific, Courtaboeuf, France) and left to dry. Harris–Schorr staining was then performed according to the WHO protocol (*World Health Organization, 2013*), and at least 150 sperm were analyzed per animal.

## Whole testis optical clearing and 3D image reconstructions
### Optical clearing (adapted from uDISCO and fast 3D clear protocols)

Adult mice were euthanized by cervical dislocation and then transcardiac perfused with 1X PBS (Sigma-Aldrich, Saint-Quentin-Fallavier, France). The testes were extracted and fixed for 2 days at 4°C in 4% PFA (Electron Microscopy Sciences, Hatfield, PA, USA). Samples were then washed with PBS for at least two days. Mouse testes were subsequently dehydrated in graded series of tert-butanol solutions (Sigma-Aldrich, Saint-Quentin-Fallavier, France) at 35°C as follows: 30% overnight (O/N), 50% for 24 hr, 70% for 10 hr, 80% O/N, 90% for 10 hr, 96% O/N, and 100% for 10 hr. The testes were cleared in clearing solution (96% Histodenz, 2% sorbitol, 20% urea) for 2 days. Then, nuclei were stained with 3.6 µM DAPI (20% DMSO; 2% Triton, 1% BSA) for 2 days. Finally, the testes were then conserved in the clearing solution at 4 °C until observation by microscopy. All these steps were carried out under agitation and protected from light.

## 3D tissue Imaging

The optically cleared mouse testes were imaged on a lightsheet fluorescence microscope (Blaze, Miltenyi Biotec, Germany), using a 4 x NA 0.35 MI PLAN objective protected by a dipping cap for organic solvents, with an overall working distance of 15 mm. Acquisitions on the horizontal plane were obtained with a fixed lightsheet thickness of 3.9 µm at both 488 and 561 nm with no overlap between horizontal planes. Voxel resolution x=1.21; y=1.21; z=2 µm. 3D reconstructions were created using Imaris software (Oxford Instruments plc, Abingdon, UK).

## Cellular image analysis

The optically cleared mouse testes were imaged using a 'ConfoBright' system which is a unique adaptive optics confocal microscope (Nikon A1R MP, Nikon Europe B.V., The Netherlands) equipped with a deformable mirror module (AOS-micro, AlpAO, Montbonnot, France) to correct geometrical aberrations. Indeed, the 3D confocal imaging of cleared sample requires long distance objectives for deep tissue imaging, resulting in optical aberrations due to inhomogeneous refractive index. The 'ConfoBright' microscope corrects these geometric optical aberrations both in excitation and detection light path and restores locally the high diffraction-limited spatial resolution and the best possible photon collection efficiency. The driving metrics based on molecular brightness were used to optimize the adaptive optics in the isoplanatic patch of ca. 100 mm at each imaged depth. Images were acquired using an apo LWD 40x/1.15 water immersion objective (WD 600 µm) and a Plan apo 10x/0.45 objective. FIJI software (open-source software *Schindelin et al., 2012*) was used to process and analyze images and Imaris software for the 3D reconstructions.

## Bioluminescence imaging

In vivo Bioluminescence imaging was performed at several time points after in vivo Luciferase -mRNA or EEV-*GFP-luciferase* injection and electroporation (*n* = 5 mice per condition). Ten minutes before imaging, mice received an intraperitoneal injection of 150 µg g$^{-1}$ of D-luciferin (Promega, Charbonnières France), and were then anesthetized (isoflurane 4% for induction and 1.5% for maintenance) before placing in the optical imaging system (IVIS Lumina III, PerkinElmer, Courtaboeuf, France). In vivo bioluminescence signals were measured in selected regions of interest (injected testes) and were expressed as mean photons per second (mean ± SEM). Background bioluminescence was measured on images from control mice. When the in vivo bioluminescence signal was no longer detectable, testes were collected and immersed in a luciferin solution for a few minutes before performing ex vivo imaging to confirm the absence of signal.

## Sperm motility

The cauda epididymis was dilacerated in 1 ml of M2 medium (Sigma-Aldrich, Saint-Quentin-Fallavier, France) and spermatozoa were allowed to swim out for 10 min at 37°C. After incubation, 30 µl of the sperm suspension was immediately placed onto analysis chamber (2X-CEL Slides, 80 µm depth, Leja Products B.V., The Netherlands) kept to 37°C for microscopic quantitative study of sperm movement. Motility of the spermatozoa was evaluated at 37°C with an Olympus microscope and Computer Aided Sperm Analysis (CASA) (CEROS II apparatus; Hamilton Thorne, Beverley, MA, USA). The settings employed for analysis were: acquisition rate: 60 Hz; number of frames: 30; minimum contrast: 30; minimum cell size: 4; low-size gate: 0.13; high-size gate: 2.43; low-intensity gate: 0.10; high-intensity gate: 1.52; minimum elongation gate: 5; maximum elongation gate: 100; magnification factor: 0.81.

The motility parameters measured were: straight line velocity (VSL); curvilinear velocity (VCL); averaged path velocity (VAP); amplitude of lateral head displacement (ALH); beat cross frequency (BCF); linearity (LIN); straightness (STR). Hyperactivated sperm were characterized by VCL > 250 µm s$^{-1}$, VSL > 30 µm s$^{-1}$, ALH > 10 µm, and LIN < 60, intermediate by VCL > 120 µm s$^{-1}$ and ALH > 10 µm, progressive sperm by VAP > 50 µm s$^{-1}$ and STR > 70% and slow sperm by VAP < 50 and VSL < 25 µm s$^{-1}$.

## Western blot

Western blotting was performed on HEK-293T cells (ATCC, Manassas, VA, USA) transfected with EEV-*Armc2* or *Armc2*-mRNA. Cells were transfected using JetPrime (101000027; Polyplus Illkirch, France) for DNA and JetMessenger (101000056; Polyplus Illkirch, France) for mRNA vectors, both according to the supplier's recommendations. After 48 hr, the cells were washed with PBS and scraped

off before centrifuging at 4°C, 1500 RPM for 5 min. The cell pellet was then resuspended in lysis buffer (87787; Thermo Scientific, Courtaboeuf, France) supplemented with an EDTA-free cocktail of protease inhibitors (11836170001; Roche, Bale, Swiss). The suspension was stirred at 4°C for 2 hr and then centrifuged at 16,000 × $g$ at 4°C for 10 min. The protein content of the supernatant was estimated with QuantiPro BCA Assay kit (Sigma-Aldrich, Saint-Quentin-Fallavier, France) before adding 5X Laemmli + 5% β-mercaptoethanol and heating at 95°C for 10 min. For the Western blot, 30 μg of proteins were deposited on a ready-made Bis-Tris gel 12% (Thermo Fisher, Courtaboeuf, France). After the transfer, the PVDF membrane was blocked with 5% milk in Tris-Buffered Saline solution containing 0.1% Tween 20 (TTBS) before immunodetection. The anti-HA antibody (11867423001; Sigma-Aldrich, Saint-Quentin-Fallavier, France) was diluted in TTBS at 1/5000. After incubation with secondary antibodies (AP136P; Sigma-Aldrich, Saint-Quentin-Fallavier, France) diluted at 1:10,000 in TTBS, binding was revealed with an enhanced chemiluminescence detection kit (1705062; Clarity Max Western ECL Substrate; Bio-Rad, Marnes-la-Coquette, France). Membranes were imaged on a ChemiDoc system (Bio-Rad, Marnes-la-Coquette, France).

## Immunofluorescence

Immunofluorescence analysis of dissociated testicular cells was performed as follows. After collection, the tunica albuginea was removed from the testes. Then the tissue was mechanically separated with 18 G needles. Once washed with PBS, the testicular cells were placed in a dissociation medium containing 1 mg collagenase type V/ml RPMI for 20 min at 37°C. After filtration (100 μm filter) and centrifugation (5 min at 200 × $g$), the pellets were resuspended in PBS before centrifugation once again. The pellet was then fixed in 1 ml PFA 4% for 5 min. Then 10 mL of ammonium acetate (0.1 M) was added. Finally, 2 ml of medium was spread on a slide. Testicular cells were permeabilized with 0.1% PBS-Triton X-100 for 20 min at room temperature. Slides were then blocked in 10% BSA with PBS-Tween 0.1% for 30 min before incubating overnight at 4°C with primary antibodies anti-rabbit *ARMC2* (1/50) (HPA053696, Sigma-Aldrich, Saint-Quentin-Fallavier, France) and anti-guinea pig tubulin (1/100) (AA345, Swiss Institute of Bioinformatics, Lausanne, Swiss) diluted in PBS-Tween 0.1–5% BSA. Slides were washed with PBS before incubating for 1 hr with secondary antibodies: anti-guinea pig (1/500) (A-11073, Thermo Fisher, Courtaboeuf, France) and anti-rabbit (1:1000) (A-11036, Thermo Fisher, Courtaboeuf, France). Samples were counterstained with DAPI and mounted with DAKO mounting media (NC2313308; Life Technology, Courtaboeuf, France). Fluorescence images were acquired under a confocal microscope (Zeiss, Jena, Germany) fitted with a 63× oil immersion objective. Images were analyzed with ZEN lite software (Zeiss, Jena, Germany).

## Intracytoplasmic sperm injection

### Collection of gametes for ICSI

*Armc2*$^{-/-}$ sperm or *Armc2*$^{-/-}$ rescued sperm were harvested by dilaceration of the cauda epididymis. They were allowed to swim for 10 min at 37°C in 1 ml of CZB.HEPES medium containing (in mM) (HEPES 20, NaCl 81.6, KCl 4.8, MgSO₄ 1.2, CaCl₂ 1.7, KH₂PO₄ 1.2, EDTA.Na₂ 0.1, Na-lactate 31, NaHCO₃ 5, Na-pyruvate 0.3, polyvinyl alcohol 0.1 mg ml$^{-1}$, phenol red 10 mg ml$^{-1}$ (0.5% (wt/vol) in DPBS), pH 7.4 KCl 125, NaCl 2.6, Na₂HPO₄ 7.8, KH₂PO₄ 1.4 and EDTA 3 (pH 7.0)). The sperm head was separated from the tail by applying multiple piezo pulses (PiezoXpert, Eppendorf, Montesson, France).

### Eggs preparation

CD1 female mice, 7 weeks old, were superovulated by IP injection of 7.5 IU pregnant mare's serum gonadotrophin (PMSG; Centravet, Dinan, France) followed by 7.5 IU HCG (Centravet, Dinan, France) 48 hr later. Eggs were collected from oviducts about 14 hr after HCG injection. Cumulus cells were removed with 0.1% hyaluronidase in M2 medium for 5–10 min. Eggs were rinsed thoroughly and kept in KSOM medium containing (in gl$^{-1}$: NaCl 5.55, KCl 0.19, KH₂PO₄ 0.05, MgSO₄ 0.05, NaHCO₃ 2.1, CaCl₂· 2H₂O 0.250, EDTA 0.004, L-glutamine 0.146, Na-lactate 1.870, Na-pyruvate 0.020, glucose 0.04, penicillin 0.05, streptomycin 0.07, BSA 1.000, NEAA 0.5% and EAA 1%.), at 15°C for at least 15 min until required for ICSI.

## ICSI procedures

ICSI was performed according to the method described by *Yoshida and Perry, 2007*. For microinjection, *Armc2*$^{-/-}$ sperm or *Armc2*$^{-/-}$-rescued motile sperm heads were separated from the tail by applying multiple piezo pulses (PiezoXpert, Eppendorf, Montesson, France). Sperm heads were introduced into the ooplasm using micromanipulators (Micromanipulator InjectMan, Eppendorf, Montesson, France) mounted on an inverted Nikon TMD microscope (Nikon, Minato-ku, Tokyo, Japan). Eggs that survived the ICSI procedure were incubated in KSOM medium at 37°C under an atmosphere of 5% $CO_2$. Pronucleus formation was checked at 6 hr after ICSI, and outcomes were scored up to the blastocyst stage.

## In vitro fertilization

WT B6D2 sperm, *Armc2*$^{-/-}$ sperm, or *Armc2*$^{-/-}$ rescued sperm were harvested by dilaceration of the cauda epididymis. They were allowed to swim in *IVF* well in capacitated media (M16 + 2% BSA) at 37°C for 10 min. Eggs were collected from mature CD1 females (6 weeks old) synchronized with 7.5 units of PMSG and 7.5 units of human chorionic gonadotrophin (hCG) prior to collection. Cumulus were incubated for 10 min in 500 µl M16 (MR-016; Sigma-Aldrich, Saint-Quentin-Fallavier, France)/1 mg l$^{-1}$ collagenase (C8176, Sigma-Aldrich). Cumulus- and zona-free eggs were collected and rinsed twice with 500 µl M16 medium. Eggs were incubated with either WT B6D2 sperm, *Armc2*$^{-/-}$ sperm, or *Armc2*$^{-/-}$ rescued sperm in capacitated medium (37°C, 5% $CO_2$) for 4 hr. After incubation, unbound sperm were washed away and eggs were incubated with KSOM at 37°C, 5% $CO_2$. Twenty-four hours after fertilization, we scored the different stages (unfertilized oocytes, aborted embryos, and 2-cell embryos as an indication of successful fertilization).

## Statistical analyses

Statistical analyses were performed using SigmaPlot (version 10; Systat Software, Inc, San Jose, CA, USA). To account for sample variability between animals, a paired *t*-test, Mann–Whitney rank sum test, one-way ANOVA, and Wilcoxon test were used. Data are displayed as mean ± SEM. p values of *≤0.05, **≤0.01, or ***≤0.001 were considered to represent statistically significant differences.

## Materials availability statement

Biological material created in this article can be obtained by contacting the corresponding authors.

## Acknowledgements

This work was supported by CNRS, INSERM, and ANR-20-CE18-0007 grant to JE. This work was supported by the Fondation pour la Recherche Médicale, grant number « ECO202006011669 » to CV. We acknowledge the MicroCell facility (GIS IBiSA, ISdV, IAB), member of the national infrastructure France-BioImaging supported by the French National Research Agency (ANR-10-INBS-04) for optical microscopy.

## Additional information

### Funding

| Funder | Grant reference number | Author |
| --- | --- | --- |
| Agence Nationale de la Recherche | ANR-20-CE18-0007 | Jessica Escoffier |
| Fondation pour la Recherche Médicale | ECO202006011669 | Charline Vilpreux |
| Agence Nationale de la Recherche | ANR-10-INBS-04 | Florence Appaix |

The funders had no role in study design, data collection, and interpretation, or the decision to submit the work for publication.

## Author contributions
Charline Vilpreux, Data curation, Formal analysis, Validation, Investigation, Visualization, Methodology; Paul Fourquin, Data curation, Formal analysis, Validation, Investigation, Visualization, Methodology, Writing – review and editing; Guillaume Martinez, Magali Court, Florence Appaix, Formal analysis, Investigation, Methodology; Jean Luc Duteyrat, Maxime Henry, Julien Vollaire, Veronique Josserand, Jacques Brocard, Investigation, Methodology; Camille Ayad, Altan Yavuz, Resources, Investigation; Geneviève Chevalier, Lisa De Macedo, Edgar Del Llano, Célia Tebbakh, Zine Eddine Kherraf, Emeline Lambert, Sekou Ahmed Conté, Zeina Wehbe, Elsa Giordani, Charles Coutton, Corinne Loeuillet, Investigation; Sofia Andrade Rebelo, carried out the experiment during the last revision; Bernard Verrier, Resources, Investigation, Writing – review and editing; Pierre F Ray, Christophe Arnoult, Writing – review and editing; Jessica Escoffier, Conceptualization, Data curation, Formal analysis, Supervision, Funding acquisition, Validation, Investigation, Visualization, Methodology, Writing – original draft, Project administration, Writing – review and editing

## Author ORCIDs
Guillaume Martinez ![ORCID] https://orcid.org/0000-0002-7572-9096
Jacques Brocard ![ORCID] https://orcid.org/0000-0002-0752-5737
Charles Coutton ![ORCID] https://orcid.org/0000-0002-8873-8098
Christophe Arnoult ![ORCID] https://orcid.org/0000-0002-3753-5901
Jessica Escoffier ![ORCID] https://orcid.org/0000-0001-8166-5845

## Ethics
All procedures involving animals were performed in accordance with French guidelines for the use of live animals in scientific research. The study protocol was approved by the local ethics committee (ComEth Grenoble #318) and received governmental authorization (ministerial agreement #38109-2022072716142778). All animals were anesthetized with ketamine/xylazine for surgical procedures, and every effort was made to minimize suffering.

Reviewer #4 (Public review): https://doi.org/10.7554/eLife.94514.4.sa1
Reviewer #5 (Public review): https://doi.org/10.7554/eLife.94514.4.sa2
Author response https://doi.org/10.7554/eLife.94514.4.sa3

---

# Additional files

## Supplementary files
MDAR checklist

## Data availability
All data generated or analyzed during this study are included in the manuscript and supporting files.

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
