## [Editor Report · eLife Assessment]

This study reports an mRNA-based strategy for restoring sperm motility in a mouse model of monogenic male infertility. The work is technically innovative and potentially **valuable**, as it demonstrates feasibility of in vivo testicular mRNA delivery without genomic integration of foreign DNA. However, although partial recovery of sperm motility is supported, the evidence for meaningful restoration of fertility remains **incomplete**, with weak IVF outcomes and difficult-to-interpret ICSI results. In addition, mechanistic questions regarding the persistence of mRNA and the specificity of germ-cell targeting remain insufficiently resolved, limiting the strength of the authors' conclusions.

---

## [Referee Report · Reviewer #4 (Public review)]

I maintain that the images in Figure 12 (new Figure 14) do not support the authors' interpretation that 2-cell embryos resulted from in vitro fertilization (IVF) of Amrc-/- rescued sperm. They are clearly not normal 2-cell embryos and instead look very much like fragmented eggs that can be seen occasionally following in vitro fertilization procedures even when that is done with wild type eggs and sperm. The only portion of current Figure 14 that has normal looking 2-cell embryos is in panel 14A4, where wild type B6D2 sperm were used. Even in that panel, there are some fragmented eggs that the authors identify as 2-cell embryos.

The authors offer the explanation that CD1 eggs fertilized by B6D2F1 hybrid male sperm do not develop beyond the 2-cell stage, citing a 2008 paper published in Biology of Reproduction by Fernandez-Goonzalez et al. I read through that paper very carefully and even had a colleague read through it in case I missed something, but that paper says nothing at all about strain incompatibilities, much less 2-cell arrest due to them. The only crosses done in that paper are CD1 eggs x CD1 sperm and B6D2 eggs x B6D2 sperm, all by intracytoplasmic sperm injection and not standard in vitro fertilization. [Note that the paper does mention performing in vitro fertilization but says nothing about how it was done or what mouse strains were used.] I even searched the literature for information regarding incompatibility between these strains and could find nothing relevant. But even if the authors are correct and there happens to be a strain incompatibility and 2-cell arrest is expected, what the authors are calling 2-cell embryos are clearly not.

A second explanation offered by the authors is that they used collagenase to remove the cumulus cells and that this may have affected the appearance of the embryos. This technique is actually used to remove both the cumulus cells and the zona pellucida and has been described as a gentler way to do so than other standard methods (hyaluronidase treatment followed by acid Tyrodes to remove the zona pellucida) (Yamatoya et al., Reprod Med Biol 2011, DOI 10.1007/s12522-011-0075-8). I think it is highly relevant to the current study that the method they used to remove cumulus cells also dissolves the zona, either partially or completely. Given that many of the eggs, fragmented eggs, and 2-cell embryos (from the WT sperm) in Figure 14A are lacking a zona pellucida, it seems very likely that many of the eggs were either zona-free or had partial zona dissolution from the start. In fact, the authors state in the Methods section that "Cumulus-free and zona-free eggs were collected..." for how IVF was done. Partial zona dissolution is standard in some protocols for performing IVF using frozen mouse sperm, which usually have much lower motility and overall efficacy than fresh sperm. In any case, it would improve transparency if the manuscript made clear somewhere other than buried in the Methods that the IVF procedure was done on eggs with partially or fully removed zonas, to allow proper interpretation.

In the rebuttal, the authors go on to state: "To provide additional functional evidence, we complemented the IVF experiments with ICSI using rescued Armc2-/- sperm and B6D2 oocytes, which allowed embryos to develop to the blastocyst stage. In these experiments, 25% of injected oocytes reached the blastocyst stage with rescued sperm compared to 13% for untreated Armc2-/- sperm (Supplementary Fig. 9) These results support the functional competence of rescued sperm and demonstrate partial recovery of fertilization ability following Armc2 mRNA electroporation."

Their conclusion that the data support partial recovery of fertilization ability following Armc2 mRNA electroporation in my opinion has no basis. This experiment was done only once, and no information is provided regarding how many eggs underwent ICSI or how many reached the blastocyst stage. The authors claim that the rescued sperm were better than the Armc2-/- sperm in producing blastocysts, but this is based on a simple percentage report of 25% vs 13% without any statistical analysis, even on the results from the single experiment presented.

Overall, the paper shows rescue of some sperm motility by the new method they use, and the new title is therefore appropriate. The authors have also dealt reasonably with many of the original concerns regarding documenting that their methodology was effective in producing protein (at least the GFP marker) in spermatogenic cells. In my view the authors have, however, not shown any indication of functional recovery over what is already known for the knockout sperm, that ICSI can support blastocyst stage embryo development. They also have not, in my view, justified the claims at the end of the abstract "These motile sperm were able to produce embryos by IVF..." and that "...mRNA electroporation can restore...partially fertilizing ability..."

---

## [Referee Report · Reviewer #5 (Public review)]

While the study presents an innovative and potentially impactful mRNA-based approach for addressing monogenic causes of male infertility, several significant weaknesses limit the strength of the authors' central conclusions.

First, the functional evidence for true fertility restoration remains incomplete. Although the authors convincingly demonstrate partial recovery of sperm motility, the downstream reproductive outcomes, particularly for IVF, are weak. Importantly, these concerns are shared by all three reviewers and the former Reviewing Editor, and to my eye they are both thoughtfully articulated and well warranted. The ICSI data show modest improvement, but this rescue is difficult to interpret.

In parallel, significant mechanistic questions persist regarding the unusually prolonged persistence of naked mRNA and reporter protein expression in germ cells, which is not fully reconciled with established mRNA and protein half-life biology and is supported largely by inference rather than by direct decay measurements.

Finally, although the authors have conducted additional cellular analyses, concerns about the extent and specificity of germ-cell targeting versus Sertoli-cell expression remain unresolved. Together, these issues do not negate the technical novelty of the work, but they do constrain the confidence with which the current dataset can support the authors' strongest therapeutic claims.

---

## [Author Response]

The following is the authors’ response to the previous reviews

**Public Reviews:**

**Reviewer #1 (Public review):**
The authors assess the effectiveness of electroporating mRNA into male germ cells to rescue the expression of proteins required for spermatogenesis progression in individuals where these proteins are mutated or depleted. To set up the methodology, they first evaluated the expression of reporter proteins in wild-type mice, which showed expression in germ cells for over two weeks. Then, they attempted to recover fertility in a model of late spermatogenesis arrest that produces immotile sperm. By electroporating the mutated protein, the authors recovered the motility of ~5% of the sperm; although the sperm regenerated was not able to produce offspring using IVF, the embryos reached the 2-cell state (in contrast to controls that did not progress past the zygote state).This is a comprehensive evaluation of the mRNA methodology with multiple strengths. First, the authors show that naked synthetic RNA, purchased from a commercial source or generated in the laboratory with simple methods, is enough to express exogenous proteins in testicular germ cells. The authors compared RNA to DNA electroporation and found that germ cells are efficiently electroporated with RNA, but not DNA. The differences between these constructs were evaluated using in vivo imaging to track the reporter signal in individual animals through time. To understand how the reporter proteins affect the results of the experiments, the authors used different reporters: two fluorescent (eGFP and mCherry) and one bioluminescent (Luciferase). Although they observed differences among reporters, in every case expression lasted for at least two weeks. The authors used a relevant system to study the therapeutic potential of RNA electroporation. The ARMC2-deficient animals have impaired sperm motility phenotype that affects only the later stages of spermatogenesis. The authors showed that sperm motility was recovered to ~5%, which is remarkable due to the small fraction of germ cells electroporated with RNA with the current protocol. The sperm motility parameters were thoroughly assessed by CASA. The 3D reconstruction of an electroporated testis using state-of-the-art methods to show the electroporated regions is compelling.The main weakness of the manuscript is that although the authors manage to recover motility in a small fraction of the sperm population, it is unclear whether the increased sperm quality is substantial to improve assisted reproduction outcomes. The authors found that the rescued sperm could be used to obtain 2-cell embryos via IVF, but no evidence for more advanced stages of embryo differentiation was provided. The motile rescued sperm was also successfully used to generate blastocyst by ICSI, but the statistical significance of the rate of blastocyst production compared to non-rescued sperm remains unclear. The title is thus an overstatement since fertility was never restored for IVF, and the mutant sperm was already able to produce blastocysts without the electroporation intervention.Overall, the authors clearly show that electroporating mRNA can improve spermatogenesis as demonstrated by the generation of motile sperm in the ARMC2 KO mouse model.

We thank the reviewer for this thoughtful and constructive comment. We agree that our study demonstrates a partial functional recovery of spermatogenesis rather than a complete restoration of fertility. Our main objective was to establish and validate a proof-of-concept approach showing that mRNA electroporation can rescue the expression of a missing or mutated protein in post-meiotic germ cells and result in the production of motile sperm.

To address the reviewer’s concern, we have the title and discussion to more accurately reflect the scope of our findings. The new title reads:

“Sperm motility in mice with oligo-astheno-teratozoospermia restored by in vivo injection and electroporation of naked mRNA”

In the manuscript, we now emphasize that while motility recovery was significant, complete fertility restoration was not achieved. We have also clarified that:

The 5% recovery in motile sperm represents a substantial improvement considering the small population of germ cells reached by the current electroporation method.

The 2-cell embryo formation observed after IVF serves as a strong indication of partial functional recovery

Finally, we now explicitly state in the Discussion that this approach should be considered a therapeutic proof-of-concept, demonstrating feasibility and potential, rather than a fully curative intervention.

**Reviewer #2 (Public review):**
The authors inject, into the rete testes, mRNA and plasmids encoding mRNAs for GFP and then ARMC2 (into infertile Armc2 KO mice) in a gene therapy approach to express exogenous proteins in male germ cells. They do show GFP epifluorescence and ARMC2 protein in KO tissues, although the evidence presented is weak. Overall, the data do not necessarily make sense given the biology of spermatogenesis and more rigorous testing of this model is required to fully support the conclusions, that gene therapy can be used to rescue male infertility.In this revision, the authors attempt to respond to the critiques from the first round of reviews. While they did address many of the minor concerns, there are still a number to be addressed. With that said, the data still do not support the conclusions of the manuscript.

We thank the reviewer for their careful and detailed assessment of our manuscript. We appreciate the concerns raised regarding mRNA stability, GFP localization, and the interpretation of spermatogenesis stages, and we have addressed these points in the manuscript and in the responses below.

(1) The authors have not satisfactorily provided an explanation for how a naked mRNA can persist and direct expression of GFP or luciferase for ~3 weeks. The most stable mRNAs in mammalian cells have half-lives of ~24-60 hours. The stability of the injected mRNAs should be evaluated and reported using cell lines. GFP protein's half-life is ~26 hours, and luciferase protein's half-life is ~2 hours.

We thank the reviewer for this important comment. The stability of mRNA-GFP was assessed by RT-QPCR in HEK cells and seminiferous tubule cells (Fig. 5). mRNA-GFP was detected for up to 60 hours in HEK cells and for up to two weeks in seminiferous tubule cells (Fig. 5A). Together, these results suggest that the long-lasting fluorescence observed in our experiments reflects a combination of transcript stability, efficient translation within germ cells and the slow protein turnover that is typical of the spermatogenic lineage.

(2) There is no convincing data shown in Figs. 1-8 that the GFP is even expressed in germ cells, which is obviously a prerequisite for the Armc2 KO rescue experiment shown in the later figures! In fact, to this reviewer the GFP appears to be in Sertoli cell cytoplasm, which spans the epithelium and surrounds germ cells - thus, it can be oft-confused with germ cells. In addition, if it is in germ cells, then the authors should be able to show, on subsequent days, that it is present in clones of germ cells that are maturing. Due to intracellular bridges, a molecule like GFP has been shown to diffuse readily and rapidly (in a matter of minutes) between adjacent germ cells. To clarify, the authors must generate single cell suspensions and immunostain for GFP using any of a number of excellent commercially-available antibodies to verify it is present in germ cells. It should also be present in sperm, if it is indeed in the germline.

We thank the reviewer for this insightful comment. To directly address the concern, we performed additional experiments to assess GFP expression in germ cells following in vivo mRNA delivery. GFP-encoding mRNA was injected and electroporated into the testes on day 0. On day 1, testes were collected, enzymatically dissociated, and the resulting seminiferous tubule cell suspensions were cultured for 12 hours. Live cells were then analyzed by fluorescence microscopy (Fig. 10).

We observed GFP expression in various germ cell types, including pachytene spermatocytes (53,4 %) (Fig 10 A-), round spermatids (25 %) (Fig 10B-E) and in elongated spermatids (11,4%) (Fig 10 C-E). The identification of these cell types was based on DAPI nuclear staining patterns, cell size fig 10 F, non-adherent characteristics, and the use of an enzymatic dissociation protocol.

Fluorescence imaging revealed strong cytoplasmic GFP signals in each of these populations, confirming efficient transfection and translation of the delivered mRNA. These results demonstrate that the in vivo injection and electroporation protocol enables effective mRNA transfection across multiple stages of spermatogenesis. These results confirm that the injected mRNA is efficiently translated in germ cells at various stages of spermatogenesis. Together, these data validate the germ cell-specific nature of the GFP signal, supporting the Armc2 KO rescue experiments.

As mentioned previously, we assessed the stability of mRNA-GFP using RT-QPCR in HEK cells and seminiferous tubule cells (see Fig. 5). mRNA-GFP was detected for up to 60 hours in HEK cells and for up to two weeks in seminiferous tubule cells. Together, these results suggest that the long-lasting fluorescence observed in our experiments reflects a combination of transcript stability and local translation within germ cells, as well as the slow protein turnover typical of the spermatogenic lineage.

Other comments:70-1 This is an incorrect interpretation of the findings from Ref 5 - that review stated there were ~2,000 testis-enriched genes, but that does not mean "the whole process involves around two thousand of genes"

We thank the reviewer for this helpful comment. We agree that our previous phrasing was imprecise. We have revised the sentence to clarify that approximately 2,000 genes show testis-enriched expression, rather than implying that the entire spermatogenic process is limited to these genes. The corrected sentence now reads:

“Spermatogenesis involves the coordinated expression of a large number of genes, with approximately 2,000 showing testis-enriched expression, about 60% of which are expressed exclusively in the testes”

74 would specify 'male':

we have now specified it as you suggested.

79-84 Are the concerns with ICSI due to the procedure itself, or the fact that it's often used when there is likely to be a genetic issue with the male whose sperm was used? This should be clarified if possible, using references from the literature, as this reviewer imagines this could be a rather contentious issue with clinicians who routinely use this procedure, even in cases where IVF would very likely have worked:

We thank the reviewer for this important comment. Concerns about ICSI outcomes indeed reflect two partly overlapping causes: the procedure itself (direct sperm injection and associated laboratory manipulations) and the clinical/genetic background of couples undergoing ICSI (especially men with severe male-factor infertility). Large reviews and meta-analyses report a small increase in some perinatal and congenital risks after ART/ICSI, but these studies conclude that it is difficult to fully disentangle procedural effects from parental factors. Importantly, genetic or epigenetic abnormalities in the male (which motivate use of ICSI) likely contribute to adverse outcomes in offspring, while some studies also suggest that ICSI-specific manipulations may alter epigenetic marks in embryos. For these reasons professional bodies recommend reserving ICSI for appropriate male-factor indications rather than as routine insemination for non-male-factor cases

We have revised the text accordingly to clarify this distinction:

“ICSI can efficiently overcome the problems faced. Nevertheless, concerns persist regarding the potential risks associated with this technique, including blastogenesis defect, cardiovascular defect, gastrointestinal defect, musculoskeletal defect, orofacial defect, leukemia, central nervous system tumors, and solid tumors [1-4]. Statistical analyses of birth records have demonstrated an elevated risk of birth defects, with a 30-40 % increased likelihood in cases involving ICSI [1-4], and a prevalence of birth defects between 1 % and 4 % [3]. It is important to note, however, that the origin of these risks remains debated. Several large epidemiological and mechanistic studies indicate that both the procedure itself (direct microinjection and in vitro manipulation) and the underlying genetic or epigenetic abnormalities often present in men requiring ICSI contribute to the observed outcomes [1, 3] [5, 6] . To overcome these drawbacks, a number of experimental strategies have been proposed to bypass ARTs and restore spermatogenesis and fertility, including gene therapy [7-10].”

199 Codon optimization improvement of mRNA stability needs a reference;

We have added the references accordingly: [11-15]

In one study using yeast transcripts, optimization improved RNA stability on the order of minutes (e.g., from ~5 minutes to ~17 minutes); is there some evidence that it could be increased dramatically to days or weeks?

We agree with the reviewer that codon optimization can enhance mRNA stability, but available evidence indicates that this effect is moderate. In *Saccharomyces cerevisiae*, Presnyak et al. (2015) [16] showed that codon optimization increased mRNA half-life from approximately 5 minutes to ~17 minutes, representing a several-fold improvement rather than a shift to days or weeks. Similar codon-dependent stabilization has been observed in mammalian systems, where transcripts enriched in optimal codons exhibit longer half-lives and enhanced translation efficiency [11]; [17]. However, these studies consistently report effects on the scale of minutes to hours. In mammalian cells, the prolonged stability of therapeutic or vaccine mRNAs—lasting for days—is primarily achieved through additional features such as optimized untranslated regions, chemical nucleotide modifications (e.g., N¹-methylpseudouridine), and protective delivery systems, rather than codon usage alone ([18]; [19]).

Other molecular optimizations that improve in vivo mRNA stability and translation include a poly(A) tail, which binds poly(A)-binding proteins to protect the transcript from 3′ exonuclease degradation and promotes ribosome recycling, and a CleanCap structure at the 5′ end, which mimics the natural Cap 1 configuration, protects against 5′ exonuclease attack, and enhances translational initiation [11-15]. Together, these modifications act synergistically to stabilize the transcript and support efficient translation.

472-3 The reported half-life of EGFP is ~36 hours - so, if the mRNA is unstable (and not measured, but certainly could be estimated by qRT-PCR detection of the transcript on subsequent days after injection) and EGFP is comparatively more stable (but still hours), how does EGFP persist for 21 days after injection of naked mRNA??

We thank the reviewer for this important comment. The stability of mRNA-GFP was assessed by RT-QPCR in HEK cells and seminiferous tubule cells (Fig. 5). mRNA-GFP was detected for up to 60 hours in HEK cells and for up to two weeks in seminiferous tubule cells (Fig. 5). Together, these results suggest that the long-lasting fluorescence observed in our experiments reflects a combination of transcript stability, efficient translation within germ cells and the slow protein turnover that is typical of the spermatogenic lineage.

Curious why the authors were unable to get anti-GFP to work in immunostaining?

We appreciate the reviewer’s question. We attempted to detect GFP using several commercially available anti-GFP antibodies under various standard immunostaining conditions. However, in our hands, these antibodies consistently produced either no signal or high background staining, making the results unreliable. We therefore relied on direct detection of GFP fluorescence, which provides a more accurate and specific readout of protein expression in our system.

In Fig. 3-4, the GFP signals are unremarkable, in that they cannot be fairly attributed to any structure or cell type - they just look like blobs; and why, in Fig. 4D-E, why does the GFP signal appear stronger at 21 days than 15 days? And why is it completely gone by 28 days? This data is unconvincing.

We would like to thank the reviewer for their comments. Figure 3–4 provides a global overview of GFP expression on the surface of the testis. The entire testis was imaged using an inverted epifluorescence microscope, and the GFP signal represents a composite of multiple seminiferous tubules across the tissue surface. Due to this whole-organ imaging approach, it is not possible to resolve individual structures such as the basement membrane or lumen, which is why the signals may appear as diffuse “blobs.”

Regarding the time-course in Figure 4D–E, the apparent increase in GFP signal at 21 days compared with 15 days likely reflects accumulation and translation of the delivered mRNA in germ cells over time, whereas the absence of signal at 28 days corresponds to the natural turnover and degradation of GFP protein and mRNA in the tissue. We hope this explanation clarifies the observed patterns of fluorescence.

If the authors did a single cell suspension, what types or percentage of cells would be GFP+? Since germ cells are not adherent in culture, a simple experiment could be done whereby a single cell suspension could be made, cultured for 4-6 hours, and non-adherent cells "shaken off" and imaged vs adherent cells. Cells could also be fixed and immunostained for GFP, which has worked in many other labs using anti-GFP.

We thank the reviewer for this insightful comment. To directly address the concern, we performed additional experiments to assess GFP expression in germ cells following in vivo mRNA delivery. GFP-encoding mRNA was injected and electroporated into the testes on day 0. On day 1, testes were collected, enzymatically dissociated, and the resulting seminiferous tubule cell suspensions were cultured for 12 hours. Live cells were then analyzed by fluorescence microscopy (Fig. 10).

We observed GFP expression in various germ cell types, including pachytene spermatocytes (53,4 %) (Fig 10 A-), round spermatids (25 %) (Fig 10B-E) and in elongated spermatids (11,4%) (Fig 10 C-E). The identification of these cell types was based on DAPI nuclear staining patterns, cell size fig 10 F, non-adherent characteristics, and the use of an enzymatic dissociation protocol.

Fluorescence imaging revealed strong cytoplasmic GFP signals in each of these populations, confirming efficient transfection and translation of the delivered mRNA. These results demonstrate that the in vivo injection and electroporation protocol enables effective mRNA transfection across multiple stages of spermatogenesis.

These results confirm that the injected mRNA is efficiently translated in germ cells at various stages of spermatogenesis. Together, these data validate the germ cell-specific nature of the GFP signal, supporting the Armc2 KO rescue experiments.

As mentioned previously, we assessed the stability of mRNA-GFP using RT-QPCR in HEK cells and seminiferous tubule cells (see Fig. 5). mRNA-GFP was detected for up to 60 hours in HEK cells and for up to two weeks in seminiferous tubule cells. Together, these results suggest that the long-lasting fluorescence observed in our experiments reflects a combination of transcript stability and local translation within germ cells, as well as the slow protein turnover typical of the spermatogenic lineage.

In Fig. 5, what is the half-life of luciferase? From this reviewer's search of the literature, it appears to be ~2-3 h in mammalian cells. With this said, how do the authors envision detectable protein for up to 20 days from a naked mRNA? The stability of the injected mRNAs should be shown in a mammalian cell line - perhaps this mRNA has an incredibly long half-life, which might help explain these results. However, even the most stable endogenous mRNAs (e.g., globin) are ~24-60 hrs.

We did not directly assess the stability of luciferase mRNA, but we evaluated the persistence of GFP mRNA, which was synthesized and optimized using the same sequence optimization and chemical modification strategy as the luciferase mRNA. In these experiments, mRNA-GFP was detectable in seminiferous tubule cells for up to two weeks after injection. We therefore expect a similar stability profile for the luciferase mRNA. These findings suggest that the prolonged fluorescence or bioluminescence observed in our study likely reflects a combination of factors, including enhanced transcript stability, local translation within germ cells, and the inherently slow protein turnover characteristic of the spermatogenic lineage.

527-8 The Sertoli cell cytoplasm is not just present along the basement membrane as stated, but also projects all the way to the lumina:

we clarified the sentence **"** Sertoli cells have an oval to elongated nucleus and the cytoplasm presents a complex shape (“tombstone” pattern) along the basement membrane, with long projections that extend toward the lumen."

529-30 This is incorrect, as round spermatids are never "localized between the spermatocytes and elongated spermatids" - if elongated spermatids are present, rounds are not - they are never coincident in the same testis section:

We thank the reviewer for this important comment and for drawing attention to the detailed staging of the seminiferous epithelium. We agree that the spatial organization of germ cells varies depending on the stage of spermatogenesis. While round spermatids (steps 1–8) and elongated spermatids (steps 9–16) are typically associated with distinct stages, transitional stages of the seminiferous epithelium can contain both cell types in close proximity, reflecting the continuous and overlapping nature of spermatid differentiation (Meistrich, 2013, Methods Mol. Biol. 927:299–307). We have revised the text to clarify this point, indicating that the relative positioning of germ cell types depends on the stage of the seminiferous cycle rather than implying their constant coexistence within the same tubule section.

Fig. 7. To this reviewer, all of the GFP appears to be in Sertoli cell cytoplasm In Figs 1-8 there is no convincing evidence presented that GFP is expressed in germ cells! In fact, it appears to be in Sertoli cells.

We thank the reviewer for their observation. As previously mentioned, we have included an additional experiment specifically demonstrating GFP expression in germ cells (fig 10). This new data provides clear evidence that the GFP signal is not restricted to Sertoli cells and confirms successful uptake and translation of GFP mRNA in germ cells.

Fig. 9 - alpha-tubuline?

We corrected the figure.

Fig. 11 - how was sperm morphology/motility not rescued on "days 3, 6, 10, 15, or 28 after surgery", but it was in some at 21 and 35? How does this make sense, given the known kinetics of male germ cell development??

We note the reviewer’s concern regarding the timing of motile sperm appearance. Variability among treated mice is expected because transfection efficiency differed between spermatogonia and spermatids. Full spermiogenesis requires ~15 days, and epididymal transit adds ~8 days, consistent with motile sperm appearing around 21 days post-injection in some mice.

And at least one of the sperm in the KO in Fig. B5 looks relatively normal, and the flagellum may be out-of-focus in the image? With only a few sperm for reviewers to see, how can we know these represent the population?

We thank the reviewer for their comment. Upon closer examination of the image, the flagellum of the spermatozoon in question is clearly abnormally short and this is not due to being out of focus. Furthermore, the supplementary figure shows that the KO consistently lacks normal spermatozoa. These defects are consistent with previous findings from our laboratory [22], confirming that the observed phenotype is representative of the KO population rather than an isolated occurrence.

**Reviewer #3 (Public review):**
Summary:The authors used a novel technique to treat male infertility. In a proof-of-concept study, the authors were able to rescue the phenotype of a knockout mouse model with immotile sperm using this technique. This could also be a promising treatment option for infertile men.Strengths:In their proof-of-concept study, the authors were able to show that the novel technique rescues the infertility phenotype of Armc2 knockout spermatozoa. In the current version of the manuscript, the authors have added data on in vitro fertilisation experiments with Armc2 mRNA-rescued sperm. The authors show that Armc2 mRNA-rescued sperm can successfully fertilise oocytes that develop to the blastocyst stage. This adds another level of reliability to the data.Weaknesses:Some minor weaknesses identified in my previous report have already been fixed. The technique is new and may not yet be fully established for all issues. Nevertheless, the data presented in this manuscript opens the way for several approaches to immotile spermatozoa to ensure successful fertilisation of oocytes and subsequent appropriate embryo development.[Editors' note: The images in Figure 12 do not support the authors' interpretation that 2-cell embryos resulted from in vitro fertilization. Instead, the cells shown appear to be fragmented, unfertilized eggs. Combined with the lack of further development, it seems highly unlikely that fertilization was successful.]

We thank the reviewer for their careful evaluation and constructive feedback. We appreciate the acknowledgment of the strengths of our study, particularly the proof-of-concept demonstration that *Armc2*-mRNA electroporation can rescue sperm motility in *Armc2* knockout mice.

Regarding the concern raised by the editor about Figure 12, we would like to clarify two technical points. First, the IVF experiments were performed using CD1 oocytes and B6D2 sperm. Due to strain-specific incompatibilities, fertilization of CD1 oocytes by B6D2 sperm typically does not progress beyond the two-cell stage (Fernández-González [23] et al., 2008, Biology of Reproduction). Therefore, the observation of two-cell embryos represents the expected limit of development in this cross and serves as a strong indication of successful fertilization, even though further development is not possible. Second, the oocytes used in these experiments were treated with collagenase to remove cumulus cells. This enzymatic treatment can sometimes affect the morphology of early embryos, which may explain why the two-cell embryos in Figure 12 appear less regular or somewhat fragmented. We also included a control showing embryos from B6D2 sperm with the same collagenase treatment on CD1 oocytes, which yielded similar appearances (Fig14 A4).

To provide additional functional evidence, we complemented the IVF experiments with ICSI using rescued *Armc2–/–* sperm and B6D2 oocytes, which allowed embryos to develop to the blastocyst stage. In these experiments, 25% of injected oocytes reached the blastocyst stage with rescued sperm compared to 13% for untreated *Armc2*–/– sperm (Supplementary Fig. 9) These results support the functional competence of rescued sperm and demonstrate partial recovery of fertilization ability following *Armc2* mRNA electroporation.

We have clarified these points in the revised Results and Discussion sections to emphasize that the IVF data indicate partial functional recovery of rescued sperm rather than full fertility restoration. These clarifications address the editor’s concern while accurately representing the technical limitations of the strain combination used in our experiments.

**Recommendations for the authors:**

**Reviewer #1 (Recommendations for the authors):**
Fig 12 and Supplementary Fig 9 are mislabeled in the text and rebuttal.

We thank the reviewer for pointing this out. We have carefully checked the manuscript and the rebuttal text, and corrected all references to Figure 12 and Supplementary Figure 9 to ensure they are accurately labeled and consistent throughout the text.

**Reviewer #3 (Recommendations for the authors):**
The contribution of the newly added authors should be clarified. All other aspects of inadequacy raised in my previous report have been adequately addressed.No further comments.

We thank the reviewer for noting this. The contributions of the newly added authors have been clarified in the Author Contributions section of the revised manuscript. All other points raised in the previous review have been addressed as indicated.

References

(1) Hansen, M., et al., Assisted reproductive technologies and the risk of birth defects--a systematic review. Hum Reprod, 2005. 20(2): p. 328-38.

(2) Halliday, J.L., et al., Increased risk of blastogenesis birth defects, arising in the first 4 weeks of pregnancy, after assisted reproductive technologies. Hum Reprod, 2010. 25(1): p. 59-65.

(3) Davies, M.J., et al., Reproductive technologies and the risk of birth defects. N Engl J Med, 2012. 366(19): p. 1803-13.

(4) Kurinczuk, J.J., M. Hansen, and C. Bower, The risk of birth defects in children born after assisted reproductive technologies. Curr Opin Obstet Gynecol, 2004. 16(3): p. 201-9.

(5) Graham, M.E., et al., Assisted reproductive technology: Short- and long-term outcomes. Dev Med Child Neurol, 2023. 65(1): p. 38-49.

(6) Palermo, G.D., et al., Intracytoplasmic sperm injection: state of the art in humans. Reproduction, 2017. 154(6): p. F93-f110.

(7) Usmani, A., et al., A non-surgical approach for male germ cell mediated gene transmission through transgenesis. Sci Rep, 2013. 3: p. 3430.

(8) Raina, A., et al., Testis mediated gene transfer: in vitro transfection in goat testis by electroporation. Gene, 2015. 554(1): p. 96-100.

(9) Michaelis, M., A. Sobczak, and J.M. Weitzel, In vivo microinjection and electroporation of mouse testis. J Vis Exp, 2014(90).

(10) Wang, L., et al., Testis electroporation coupled with autophagy inhibitor to treat non-obstructive azoospermia. Mol Ther Nucleic Acids, 2022. 30: p. 451-464.

(11) Wu, Q., et al., Translation affects mRNA stability in a codon-dependent manner in human cells. eLife, 2019. 8: p. e45396.

(12) Gallie, D.R., The cap and poly(A) tail function synergistically to regulate mRNA translational efficiency. Genes & Development, 1991. 5(11): p. 2108-2116.

(13) Henderson, J.M., et al., Cap 1 messenger RNA synthesis with co-transcriptional CleanCap analog improves protein expression in mammalian cells. Nucleic Acids Research, 2021. 49(8): p. e42.

(14) Stepinski, J., et al., Synthesis and properties of mRNAs containing novel “anti-reverse” cap analogs. RNA, 2001. 7(10): p. 1486-1495.

(15) Sachs, A.B., P. Sarnow, and M.W. Hentze, Starting at the beginning, middle, and end: translation initiation in eukaryotes. Cell, 1997. 89(6): p. 831-838.

(16) Presnyak, V., et al., Codon optimality is a major determinant of mRNA stability. Cell, 2015. 160(6): p. 1111-24.

(17) Cao, D., et al., Unlock the sustained therapeutic efficacy of mRNA. J Control Release, 2025. 383: p. 113837.

(18) Karikó, K., et al., Incorporation of pseudouridine into mRNA yields superior nonimmunogenic vector with increased translational capacity and biological stability. Mol Ther, 2008. 16(11): p. 1833-40.

(19) Pardi, N., et al., mRNA vaccines — a new era in vaccinology. Nature Reviews Drug Discovery, 2018. 17(4): p. 261-279.

(20) Meistrich, M.L. and R.A. Hess, Assessment of Spermatogenesis Through Staging of Seminiferous Tubules, in Spermatogenesis: Methods and Protocols, D.T. Carrell and K.I. Aston, Editors. 2013, Humana Press: Totowa, NJ. p. 299-307.

(21) Au - Mäkelä, J.-A., et al., JoVE, 2020(164): p. e61800.

(22) Coutton, C., et al., Bi-allelic Mutations in ARMC2 Lead to Severe Astheno-Teratozoospermia Due to Sperm Flagellum Malformations in Humans and Mice. Am J Hum Genet, 2019. 104(2): p. 331-340.

(23) Fernández-Gonzalez, R., et al., Long-term effects of mouse intracytoplasmic sperm injection with DNA-fragmented sperm on health and behavior of adult offspring. Biol Reprod, 2008. 78(4): p. 761-72.